# Speech-to-LaTeX:
# New Models and Datasets for
# Converting Spoken Equations and Sentences

**Dmitrii Korzh**[1,2], **Dmitrii Tarasov**[3,4], **Artyom Iudin**[2]
**Elvir Karimov**[2,5], **Matvey Skripkin**[3,5], **Nikita Kuzmin**[2,5],
**Andrey Kuznetsov**[3,6], **Oleg Y. Rogov**[1,2,5], **Ivan Oseledets**[1,7]

[1] AXXX, Moscow, Russia
[2] MTUCI, Moscow, Russia
[3] FusionBrain Lab, AXXX, Moscow, Russia
[4] HSE University, Moscow, Russia
[5] Applied AI Institute, Moscow, Russia
[6] Innopolis University, Innopolis, Russia
[7] Moscow State University, Moscow, Russia

## Abstract

Conversion of spoken mathematical expressions is a challenging task that involves transcribing speech into a strictly structured symbolic representation while addressing the ambiguity inherent in the pronunciation of equations. Although significant progress has been achieved in automatic speech recognition (ASR) and language models (LM), the problem of converting spoken mathematics into LaTeX remains underexplored. This task directly applies to educational and research domains, such as lecture transcription or note creation. Based on ASR post-correction, prior work requires 2 transcriptions, focuses only on isolated equations, has a limited test set, and provides neither training data nor multilingual coverage. To address these issues, we present the first fully open-source large-scale dataset, comprising over 66,000 human-annotated audio samples of mathematical equations and sentences in English and Russian, drawn from diverse scientific domains. In addition to the ASR post-correction models and few-shot prompting, we apply audio language models, demonstrating comparable character error rate (CER) results on the MathSpeech benchmark (28% vs. 30%) for the equations conversion. In contrast, on the proposed S2L-equations benchmark, our models outperform the MathSpeech model by a substantial margin of more than 36 percentage points, even after accounting for LaTeX formatting artifacts (27% vs. 64%). We establish the first benchmark for mathematical sentence recognition (S2L-sentences) and achieve an equation CER of 40%. This work lays the groundwork for future advances in multimodal AI, with a particular focus on mathematical content recognition.

## 1 Introduction

Modern speech recognition models (Baevski et al., 2020; Radford et al., 2023) demonstrate strong performance on general speech but struggle with domain-specific tasks such as converting spoken mathematical expressions and sentences into formal symbolic representations like LaTeX. While simple symbols (e.g., $+$, $-$, $\pi$, $\sqrt{}$) are often correctly recognized, more complex or nested expressions remain challenging. This limitation is critical in academic and educational contexts, including automatic lecture transcription, multimodal assistant development, and scientific note-taking. Speech-to-LaTeX (S2L) models, which can interpret the structure and semantics of mathematical language in speech, are essential in these applications. Prior work, such as MathBridge (Jung et al., 2024), addresses the Text-to-LaTeX task using language models trained on textual representations of spoken equations.

Figure 1: S2L methods schematic illustration. (a) Post-correction approach. (b) Multi-modal end-to-end approach (SALMONN). In (a), audio is transcribed by an ASR model, and the result is passed to an LLM for LaTeX conversion. In (b), raw audio is processed by 2 audio encoders and an adapter, and the resulting audio and textual prompt tokens are fed into a LLaMA-based LLM to generate the LaTeX.

The S2L problem, however, remains largely underexplored. MathSpeech (Hyeon et al., 2025b) proposes an ASR post-correction approach that transcribes spoken equations into text, followed by Text-to-LaTeX generation via language models. Evaluation was performed on 1.1k spoken expressions from YouTube. However, this pipeline depends on dual ASR transcriptions, supports only isolated equations (not mathematical sentences), lacks multilingual support, and omits end-to-end multimodal approaches. Moreover, the test set is limited in size and diversity, and the underlying training data, voiced-over with TTS from `MathBridge` equations and pronunciations, is not publicly released. To address these limitations, it is necessary to develop new S2L datasets that contain partial human annotations and employ more robust modeling techniques, including end-to-end systems that integrate ASR, LMs, and audio-based LLMs.

This paper introduces the `S2L` dataset for spoken mathematical language, which consists of 2 subsets: `S2L-sentences` and `S2L-equations`. The dataset contains approximately 12k unique mathematical sentences and 10.7k distinct isolated equations, each annotated by up to 3 different speakers (from a total of 33 annotators) to capture diverse pronunciations, intonations, and linguistic styles. To further expand and augment the dataset, additional artificially-annotated expressions and sentences were added, resulting in 571k generated audio samples.

We develop several S2L methods combining state-of-the-art ASR models (Radford et al., 2023; Chen et al., 2022) with post-processing via fine-tuned LMs and end-to-end approaches, based on Audio-LLMs (Tang et al., 2024; Chu et al., 2023). These approaches are illustrated in Figure 1. Our best models achieve an equation CER between 27.7% and 30.0% on English data. Within mathematical sentences, text CER is up to 9.6%, and equation CER is up to 39.7%. These relatively high rates reflect the inherent ambiguity in spoken math. For example, "kappa" may correspond to \kappa ($\kappa$) or \varkappa ($\varkappa$); the phrase "one over x plus two" could yield $\frac{1}{x} + 2$, $\frac{1}{x+2}$, or $1/x + 2$. Despite such ambiguities, our models generate valid LaTeX in most cases, establishing a strong performance baseline.

**Our contributions might be summarized as follows:**

- We release[1] the first large-scale, open-source dataset of spoken mathematical expressions and sentences (`S2L-sentences`, `S2L-equations`) in English and Russian, including 66k human and 571k synthetic audio samples with diverse pronunciations and complexities.

- We evaluate multiple S2L methods based on ASR post-correction, few-shot prompting, and audio-LLM integration, demonstrating strong performance across metrics and outperforming MathSpeech on several tasks.

- We conduct a comprehensive evaluation using relevant metrics to establish robust baselines and detailed analysis for future S2L research[2].

---

[1] `https://huggingface.co/datasets/marsianin500/Speech2Latex`.
[2] Code available at `https://github.com/dkorzh10/speech2latex`.

## 2    RELATED WORK

**Automatic Speech Recognition.** CTC loss (Graves et al., 2006; Amodei et al., 2016) allows alignment between audio and text without a precise labelling. While traditional ASR suffers from context insensitivity, the Conformer model (Gulati et al., 2020) combines convolutions and self-attention to capture both local and global dependencies. Wav2Vec 2.0 (Baevski et al., 2020) uses contrastive self-supervised learning to extract high-quality audio features. Whisper (Radford et al., 2023), a transformer-based architecture, is trained in a weakly-supervised manner, demonstrating a robust performance across various audio domains.

**Language Models.** Transformer-based LMs such as BERT (Devlin, 2018), T5 (Raffel et al., 2020), and GPT-3 (Radford et al., 2019) have shown strong performance, including in math-related problems. LMs can process structured and ambiguous data, such as chemistry formulas (Ganeeva et al., 2024) or code problems (Li et al., 2024). Recent models such as Qwen2.5-Math (Yang et al., 2024) and InternLM-Math (Ying et al., 2024) are tuned explicitly for mathematical reasoning, leveraging chain-of-thought prompting and large-scale math corpora. However, they still require fine-tuning for the text-to-LaTeX conversion.

**ASR Post-Correction.** Post-correction pipelines are well studied (Ma et al., 2025; 2023; Chen et al., 2023) and effective due to the availability of textual and especially textual math data (e.g., `MathBridge`), compared to available audio data used for fine-tuning ASR models. These approaches use ASR to transcribe audio and then apply an LM to convert the text into LaTeX. This approach leverages strong LLM priors, which might be pre-trained on relevant mathematical data, without requiring expensive audio annotation. However, performance heavily depends on transcription quality, and ambiguity in mathematical speech remains challenging.

**Audio-LLMs.** Multimodal LLMs (M-LLMs) aim to jointly process audio and text by encoding multiple modalities and feeding them into a unified LM. SALMONN (Tang et al., 2024) combines Whisper and BEATs embeddings via Q-former (Li et al., 2023) into a LLaMA-based decoder, enabling tasks like ASR and audio QA. Qwen-Audio adapts Whisper encodings to handle instruction-based audio tasks across multiple languages. While promising, these models are not explicitly designed for mathematical LaTeX generation, lack fine control over symbolic precision, and often cannot recognize spoken mathematics completely.

**OCR LaTeX Recognition.** In contrast to S2L, OCR-based LaTeX recognition has received significant attention (Genthial, 2024; Blecher et al., 2023; OleehyO, 2024).

**Text-to-Speech (TTS).** Modern TTS models (Casanova et al., 2024; Kong et al., 2020) can synthesize natural speech at near-human quality. Recently, MathReader (Hyeon et al., 2025a) proposed a pipeline for converting LaTeX to speech via LLM-generated pronunciation and standard TTS. Authors of (Roychowdhury et al., 2025) evaluated 5 TTS systems on math expressions, showing that performance varies significantly by expression category and performs substantially worse than human expert readings.

**Spoken Mathematics Recognition.** Only a few works tackle S2L-related problems directly. Mathifier (Batlouni et al., 2011) targeted fixed-template equation recognition, now largely outdated. The work (Wei et al., 2025) introduced the `Spoken-MQA` benchmark for spoken math reasoning and evaluated several ASR post-correction models and audio-LLMs. While they demonstrated promising results on arithmetic reasoning, LaTeX-style symbolic expressions and advanced expressions were largely absent.

MathSpeech (Hyeon et al., 2025b) introduced a post-correction pipeline using 2 ASR transcripts as input. Equations from `MathBridge` were synthesized via TTS and transcribed with 4 ASR models to produce around 8M samples. 2 T5-small models were trained to correct and convert transcripts into LaTeX. While effective, their approach requires multiple ASRs, lacks sentence-level context, and does not support multilingual or end-to-end modeling.

**Datasets.** Textual math datasets like `Proof-Pile` (Azerbayev et al., 2023; Weber et al., 2024) and `OpenWebMath` (Paster et al., 2023) are key for training math-aware LLMs. `MathBridge` provides 23M LaTeX expressions with artificial text and context, but suffers from low quality and duplicates. OCR-LaTeX datasets like `TextTeller` (OleehyO, 2024) offer high-quality image-LaTeX pairs and can potentially support S2L via voice-over. However, large-scale S2L datasets

remain missing: `MathSpeech` offers only a 1.1k test set with no training data, and `Spoken-MQA` contains just 2.3k TTS samples focused on basic arithmetic. No dataset provides large-scale, human-annotated, contextual spoken math data, motivating us to begin with dataset collection. A quality benchmark and protocol for LaTeX document generation are introduced in (Kale & Nadadur, 2025).

# 3 DATASET COLLECTION

In this section, we describe the pipeline of `S2L` data collection. We combined human-annotated and artificially generated data to create a robust and diverse dataset. We began by collecting mathematical equations and sentences from multiple sources, along with corresponding reference pronunciations. These pronunciations serve both as guidance for non-expert human annotators and as required inputs for artificial annotation via TTS or voice-conversion (VC) models.

Each sample is classified by language (English or Russian), annotation type (human or artificial), source (e.g., `Proof-Pile`, `MathBridge`, `TextTeller`, or generated), and format (`S2L-equations` for isolated expressions vs. `S2L-sentences` for in-context mathematical sentences).

## 3.1 DATA SOURCES AND PREPARATION

For the `S2L-equations`, we utilized two existing sources, `MathBridge`, and `TextTeller`, and also generated additional equations. For the `S2L-sentences`, the primary source was `Proof-Pile`.

We incorporated a subset of `MathBridge`, which offers large-scale textual pairs of equations and pronunciations with surrounding context. However, `MathBridge` data quality is inconsistent. Common issues include: (i) text instead of a formula; (ii) invalid LaTeX; (iii) missing pronunciations; (iv) duplicated entries; (v) pronunciation additionally contains LaTeX; (vi) mismatched formula–pronunciation pairs; (vii) lots of nearly duplicated formulas, such as $\cos(\alpha), \ldots, \cos(\omega)$. We selected 15,000 candidate equations and filtered them manually, retaining 3,000 high-quality English samples for both human and artificial annotation. Furthermore, we employed heuristic filters and LaTeX compilation checks to automatically clean the whole dataset, reducing it from 23 million to 1.5 million validated samples; of these, 400k were subsequently annotated with TTS in English. The heuristics involved filtering out overly short formulas, text-only entries (which typically contained plain text rather than equations), and cases where the equation was substantially longer than its spoken form.

`TextTeller` provides complex equations used in OCR-LaTeX research. We extracted 9,400 unique LaTeX equations and used GPT-4 to generate 4 reference pronunciations (2 English, 2 Russian) per equation, later used for artificial voice synthesis.

To enhance diversity across mathematical domains, we prompted GPT-4 to generate LaTeX–pronunciation pairs for common study topics (e.g., Calculus, Mechanics). Examples of generated topics and corresponding equations are provided in Appendix A.1.

For `S2L-sentences`, we extracted contextual math sentences from the `arxiv` subset of `Proof-Pile-2`. Next preprocessing steps were applied: (i) filtering for inline formulas; (ii) removing LaTeX formatting of text (e.g., \citep, \textit); and (iii) validating equations via KaTeX compilation. Sentences were stratified by equation length (Table 6) and balanced accordingly, resulting in 12.4k clean samples with a human-annotation coverage rate of approximately 2. Additionally, we included 1.4k negative examples (no equations) from the `LRS3` dataset (Afouras et al., 2018) for artificial annotation.

84% equations from `S2L-equations` primarily have length between 3 and 50 symbols, while the rest are primary up 140 symbols and the max length is 230. The majority of samples from `S2L-sentences` have 1-4 equations per sentence, while the max is 11. The detailed statistics and stratification by equation length and number of equations per sentence are presented in Tables 6, 7 in the Appendix due to the limited space.

## 3.2 EQUATIONS NORMALIZATION

All LaTeX equations were normalized using a KaTeX (Barabash et al., 2025) fork, with uncompilable samples removed. The process standardized notation, eliminated extraneous spaces, inserted required braces, and unified operator names by parsing and reconstructing formulas via Abstract Syntax Tree. This reduced the CER by 1% on `S2L-Equations`. See Table 1 for examples.

Table 1: Examples of LaTeX Equation Normalization.

| Original Equation | Normalized Equation |
|---|---|
| `\sum_i^n i` | `\sum_{i}^{n}i` |
| `\frac{ n( n+1 ) }{ 2 }` | `\frac{n(n+1)}{2}` |
| `\underset{ \xi }{ \max }` | `\max_{\xi}` |
| `\Delta z\sim1` | `\Delta\ z\sim\ 1` |

## 3.3 DATASET COMPOSITIONS AND AUDIO ANNOTATION

Each distinct equation or sentence in our dataset is paired with at least one reference pronunciation, which serves both as a TTS input and a human reference. For augmentation, multiple pronunciations were collected for several thousand expressions. To reduce annotation cost and augment the data, we explored the viability of training models on artificially generated audio. For this, we used open-source XTTSv2 (Casanova et al., 2024), and proprietary TTS APIs (e.g., SaluteSpeech). XTTSv2 was selected as the primary annotator due to its public availability, high audio fidelity, and voice conversion capability. For human annotation, we used a crowd-sourcing platform similar to MTurk. The process showed speakers a formula or sentence and reference pronunciations. Overall, 33 unique human annotators were involved. Manual verification was performed for each annotator: 10% of their audio was reviewed, and if more than 15% was rejected due to noise or low quality, only the verified subset was retained. **To sum up, the resulting statistics are the following:**

- Human annotation
  - `S2L-equations` (Eng): 6,535 unique equations, 27 annotators, total 23,196 audio.
  - `S2L-equations` (Rus): 4,274 unique equations, 10 annotators, total 18,134 audio.
  - `S2L-sentences` (Eng): 12,395 unique sentences, 20 annotators, total 24,794 audio.
- Artificial annotation
  - `S2L-equations` (Eng): 406,122 unique equations (6,535 as for human annotators, 399,587 new), 9 artificial voices, total 450,874 audio.
  - `S2L-equations` (Rus): 12,669 unique equations (4,274 as for human annotation, 8,395 new), 14,449 reference pronunciations, 8 artificial voices, total 53,109 audio.
  - `S2L-sentences` (Eng): 12,064 (10,411 as in human annotation, 1,984 new) unique sentences, 4 artificial voices, total 67,069 audio.

## 3.4 S2L DATA REPRESENTATIVENESS DISCUSSION

Assessing dataset representativeness in the context of spoken mathematical expressions is inherently challenging due to the breadth of mathematical fields and the diversity of pronunciations. For example, the spoken phrase "2 squared from x plus 1" can map to either $\frac{2}{x^2+1}$ or $\frac{2}{x^2}+1$. One strategy is to explicitly include "parentheses" in the pronunciation. Some samples adopt this, but many do not, reflecting real-world variability. Our dataset does not aim to cover the full range of scientific disciplines. It rather prioritizes diversity in structure, notation, and linguistic realization. To this end, we were motivated by the following:

- Symbol and syntax coverage: We ensured broad coverage of commonly used LaTeX symbols and structures, such as `\alpha, \omega, \frac{}{}, \sqrt{}, \left(`, etc.
- Curricular diversity: We prompted GPT to generate equations and pronunciations across typical undergraduate mathematics and physics topics, removing overly simplistic samples and those dominated by textual content (e.g., `\text{}`).

- Source variability: The dataset draws from 3 distinct formula sources for `S2L-equations`, and includes real-world academic content via `TextTeller` and `Proof-Pile-2`.

- Language and voice variation: Pronunciations were collected in both English and Russian, using multiple TTS voices as well as human annotators.

- Pronunciation style diversity: In some cases, we included both phonetic (e.g., "f equals m a") and lexical (e.g., "Newton's second law") variants, reflecting natural pronunciations.

While GPT-based samples may introduce some bias in both equations and pronunciations, this is partially mitigated through the inclusion of real-world academic data and crowd-sourced spoken annotations. Empirically, we observe that models trained on synthetically generated audio generalize well to human-annotated test cases, with no drastic degradation in performance (see the next section).

Table 2: `S2L-Equations` results. Disjoint split: test equations do not overlap with train equations. "A": artificially (TTS) annotated audio except 400k samples extracted from `MathBridge`; "H": human-annotated audio; "Mix": combination of "A" and "H"; CER is calculated for lower-case. "Q-$\alpha$B" and "Q-math-$\alpha$B" stand for Qwen2.5-$\alpha$B-instruct and Qwen2.5-math-$\alpha$B-instruct, respectively. "Full" implies addition of 400k artificially annotated samples from `MathBridge` to the "A" set.

| Model | Train | Train | Test | Test: Mix | | Test: H | | Test: A | |
|---|---|---|---|---|---|---|---|---|---|
| | | Language | Language | CER | TeXBLEU | CER | TeXBLEU | CER | TeXBLEU |
| MathSpeech | MS-train | Eng | Eng | 64.04 | 83.71 | 59.32 | 83.64 | 69.65 | 83.80 |
| Q-0.5B | A | Eng | Eng | 33.28 | 88.61 | 33.26 | 88.54 | 33.30 | 88.70 |
| Q-0.5B | A | Eng+Rus | Eng | 34.78 | 87.90 | 34.94 | 87.57 | 34.59 | 88.31 |
| Q-0.5B | H | Eng | Eng | 36.91 | 87.86 | 35.01 | 88.25 | 39.16 | 87.38 |
| Q-0.5B | H | Eng+Rus | Eng | 35.43 | 88.06 | 33.94 | 88.47 | 37.19 | 87.56 |
| Q-0.5B | Mix | Eng | Eng | 31.41 | 88.83 | 31.06 | 88.87 | 31.82 | 88.78 |
| Q-0.5B | Mix | Eng+Rus | Eng | 32.33 | 88.60 | 31.18 | 88.89 | 33.69 | 88.24 |
| Q-0.5B | Mix-full | Eng+Rus | Eng | 27.21 | 90.20 | 27.03 | 90.14 | 27.42 | 90.27 |
| Q-1.5B | A | Eng | Eng | 31.24 | 89.22 | 31.37 | 89.15 | 31.08 | 89.31 |
| Q-1.5B | A | Eng+Rus | Eng | 30.73 | 88.92 | 30.70 | 88.73 | 30.77 | 89.16 |
| Q-1.5B | H | Eng | Eng | 29.69 | 89.41 | 27.57 | 89.69 | 32.22 | 89.07 |
| Q-1.5B | H | Eng+Rus | Eng | 30.93 | 89.04 | 28.85 | 89.42 | 33.39 | 88.57 |
| Q-1.5B | Mix | Eng | Eng | 29.76 | 89.28 | 28.93 | 89.44 | 30.74 | 89.09 |
| Q-1.5B | Mix | Eng+Rus | Eng | 31.14 | 89.37 | 30.08 | 89.43 | 32.40 | 89.28 |
| Q-1.5B | Mix-full | Eng+Rus | Eng | 25.69 | 90.70 | 24.91 | 90.74 | 26.61 | 90.66 |
| Q-Math-1.5B | A | Eng+Rus | Eng | 29.57 | 90.00 | 29.44 | 89.80 | 29.74 | 90.23 |
| Q-Math-1.5B | H | Eng+Rus | Eng | 31.45 | 89.25 | 30.71 | 89.43 | 32.34 | 89.02 |
| SALMONN-13B | Mix-full | Eng | Eng | 17.50 | 93.68 | 18.17 | 93.64 | 16.70 | 93.72 |
| Gemma-3n-8B | Mix-full | Eng | Eng | 34.24 | 89.15 | 33.24 | 89.23 | 35.42 | 89.06 |
| Flamingo-3-8B | Mix | Eng | Eng | 23.25 | 91.32 | 23.13 | 91.31 | 23.40 | 91.33 |

## 4 METHODOLOGY AND EXPERIMENTAL SETUPS

Training hyperparameters are described in the Appendix B due to the limited space. Prior to training, all audio was resampled to 16kHz to ensure consistency across ASR and Audio-LLM pipelines, and `$` signs were additionally excluded from `S2l-equations` labels.

### 4.1 ASR POST-CORRECTION

We first evaluated a Whisper-Large v3 ASR-only baseline for LaTeX transcription, achieving 88% CER on English `S2L-equations` - deemed insufficient. While shallow/deep fusion methods could improve performance, we excluded them due to practical limitations: high memory/latency costs (shallow fusion) and training complexity (deep fusion). Instead, we adopted an ASR post-correction pipeline, a strategy previously shown to be effective for transcription improvement and for math-related speech tasks (Hyeon et al., 2025b; Jung et al., 2024; Chen et al., 2023). Among the ASR models evaluated, Whisper-Large v3 provided the most accurate transcriptions for mathematical symbols, particularly Greek letters and structured expressions. Canary and Qwen-Audio (based on Whisper v2) also performed reasonably well, while WavLM and Wav2Vec2.0 produced frequent symbol errors. Please, refer to Table 8 in the Appendix for the transcriptions' comparison. Summing up, we used a frozen ASR model and fine-tuned LLMs for the post-correction.

Table 3: SALMONN-13B prediction examples on `S2L-equations`, test subset.

| Prediction | Ground Truth | CER, % | Pronunciation |
|---|---|---|---|
| $F_{\mu\nu} = \partial_\mu A_\nu - \partial_\nu A_\mu$ | $F_{\mu\nu} = \partial_\mu A_\nu - \partial_\nu A_\mu$ | 0.0 | The field strength tensor for electromagnetism is F mu nu equals d mu A nu minus d nu A mu. |
| $E = \frac{F}{q}$ | $\mathbf{E} = \frac{\mathbf{F}}{q}$ | 54.5 | An electric field equals force over charge. |
| $n(\mu, \sigma^2, t)$ | $\mathcal{N}(\mu, \frac{\sigma^2}{T})$ | 57.4 | N of mu, sigma squared over T. |
| $\text{Var}(X) = r\frac{1-p}{p^2}$ | $\text{Var}(X) = \frac{r(1-p)}{p^2}$ | 13.9 | For a negative binomial distribution, the variance equals R times 1 minus p divided by p squared. |
| $n(\gamma, \theta_e)/n = \delta(\theta_e - \theta_j)$ | $n(\Gamma, \theta_e)/n = \delta(\theta_e - \theta_j)$ | 1.1 | N of Gamma, theta sub e divided by N equals delta of theta sub e minus theta sub j. |
| $\text{Ei}(x) = \frac{1}{\pi} \int_0^\infty \cos\left(\frac{t^3}{3} + xt\right) dt$ | $\text{Ai}(x) = \frac{1}{\pi} \int_0^\infty \cos\left(\frac{t^3}{3} + xt\right) dt$ | 0.6 | Airy function of the first kind, ai of x is equal to one divided by pi times the integral from zero to infinity of cosine of t cubed divided by three plus x times t dt. |
| $\lim_{x \to -5} \frac{\sqrt{4-x-3}}{x+5}$ | $\lim_{x \to -5} \frac{\sqrt{4-x}-3}{x+5}$ | 10.0 | Limit as x tends to negative 5 of square root of 4 minus x minus 3 divided by x plus 5. |
| $\sum_{i=1}^n i \cdot i = \frac{n(n+1)(2n+1)}{6}$ | $\sum_{i=1}^n i \cdot i = \frac{n(n+1)(2n+1)}{6}$ | 0.0 | The sum from i equals one to n of i times i equals n times n plus one times two n plus one divided by six. |
| $1 \leq u_1, u_2, b_1, v_2 \leq d$ | $1 \leq u_1, u_2, v_1, v_2 \leq d$ | 1.3 | 1 is less than or equal to u sub 1, u sub 2, v sub 1, v sub 2, which are all less than or equal to d. |

For `S2L-equations`, experiments were conducted using Qwen2.5 and Qwen2.5-Math across English, Russian, and combined splits. All LLMs received the ASR transcription as input and produced LaTeX equations or sentences as output. For `S2L-sentences`, we fine-tuned the Qwen2.5 (0.5B, 1.5B, and 7B) and Qwen2.5-Math-1.5 instruct models using 3 training splits: artificial, human-annotated, and mixed. To assess few-shot performance, we tested the same models using a 5-shot prompt format, evaluating generalization across parameter sizes.

## 4.2 MULTIMODAL MODELS

A multimodal S2L pipeline was further explored using Audio-LLMs. This approach bypasses phonetic transcription by directly converting raw audio into LaTeX expressions or sentences. Audio encoders first extract latent features from the waveform; these are then processed by a modality adapter to align with LLM token embeddings. The resulting audio tokens are concatenated with the textual prompt's tokens and passed to the LLM for decoding.

We used Qwen-Audio, Gemma-3n (Team et al., 2025), Audio Flamingo-3 (Goel et al., 2025), and SALMONN-13B for this setting, given their strong benchmark performance. LLMs were fine-tuned using the LoRA technique (Hu et al., 2022), while freezing audio encoders and the adapter. Since the Qwen-Audio fine-tuning pipeline was not publicly available, we prepared it ourselves.

## 4.3 EVALUATION

The primary reported metrics are character error rate (CER), and TeXBLEU (Jung et al., 2025) metric, recently specifically proposed for LaTeX comparison. There are cases where semantically equivalent LaTeX formulas differ in syntax, which can distort formal metrics based on raw code. For instance, the expressions `\int_{a}^{b} f(x) dx` and `\int_a^bf(x)dx` represent the same formula but have a high CER. Additionally, capitalization (e.g., `\phi` vs. `\Phi`) and font styles (e.g., `\mathcal{R}` vs. `r`) introduce further ambiguities. To mitigate these effects, we apply equation normalization as previously described in subsection 3.2. Additionally, all metrics are evaluated on lowercase text except for TeXBLEU.

Table 4: Comparison with MathSpeech on the `MathSpeech` benchmark and `S2L-equations` (English test). Metric: CER. Qwen: Qwen2.5-0.5B-Instruct (multilingual). SALMONN was tuned only in English.

| Model | MathSpeech | S2L-equations |
|---|---|---|
| MathSpeech | **27.7**% | 64.0% |
| Qwen | 30.0% | 27.2% |
| SALMONN | **27.7**% | **17.5**% |

For `S2L-equations`, predictions and ground truth are compared in LaTeX form, as illustrated in Table 3. For `S2L-sentences`, which contain inline formulas within English text, we separately evaluate both equation and text components. All formulas are extracted from the predicted sequences, concatenated, and compared against the reference formulas using character-level metrics.

## 4.4 DATASET SPLITS

We explored several dataset splitting strategies to evaluate generalization under different conditions. (i) **disjoint formula split**: in this setup, the train and test sets contain entirely non-overlapping formulas (or sentences), ensuring that no equation seen during training appears in the test split. This setting measures the model's ability to generalize beyond memorization of specific formulas; (ii) **source-type split**: to assess the utility of synthetic data, we constructed splits where the test set consists solely of human-annotated audio, while the training set comprises either synthetic (TTS), human, or mixed audio. This configuration evaluates whether models trained on inexpensive artificial speech can generalize to real human input; (iii) **monolingual vs. bilingual**: we examined the effect of cross-lingual data by comparing monolingual and bilingual training setups. This analysis tests whether training on both English and Russian subparts improves generalization, or whether language-specific models perform better on a test of a fixed language.

## 5 RESULTS AND DISCUSSION

### 5.1 S2L-EQUATIONS RESULTS

Table 2 compares the performance of post-ASR and multimodal S2L models on the English `S2L-equations` test subset. Due to the limited space, the complete Table 10 with the Russian test and additional splits is moved to the Appendix. The key observations are the following:

- Multilingual training is not always beneficial. For Qwen2.5-0.5B, Human English Test scores are worsened 33.94% vs. 35.01% with Eng vs. Eng+Rus training, while for Qwen2.5-Math they improved from 30.71% to 28.08%.

- For English experiments, adding 400k TTS samples improves metrics, but for the Russian test results worsen, likely due to language imbalance.

- SALMONN achieves superior results over all models. Flamingo-3 performs only on par with smaller post-correction LMs. Gemma and Qwen-Audio perform below even small post-correction LMs.

- 1.5B models outperform 0.5B, but 7B model does not always outperform 1.5B and 0.5B models, likely because they were trained with LoRA and frozen weights, unlike the fully fine-tuned smaller models.

- Math-oriented Qwen2.5-Math-1.5B shows no clear advantage over Qwen2.5-1.5B, likely because inputs are given as natural language rather than mathematical expressions.

We conducted experiments by adding frequently used LaTeX symbols, such as {, }, ^, _, as additional tokens not presented in a default tokenizer separately. However, this modification did not result in any measurable improvement in model performance. The KaTeX compilation success rate of predicted equations varied from 98% to 99.5%, and failure cases mainly included bracket issues. In summary, despite a large nominal CER, the metrics do not reflect the actual situation due to the considerable ambiguity of possible pronunciation and transcriptions, and our models demonstrate satisfactory quality. For instance, one can assess the generation quality of SALMONN in Table 3.

**Comparison with MathSpeech.** We evaluated our approaches on the `MathSpeech` benchmark and the MathSpeech model on our `S2L-equations` test. MathSpeech model tends to use operators that do not affect the semantics of the equation, like `\displaystyle`, `\operatorname`. In contrast, our dataset lacks them, and consequently, our models also lack them. Thus, for a fairer evaluation, we additionally normalized predictions and labels of our and MathSpeech models and datasets. This "improved" MathSpeech's metric on `S2L-equations` from 92% to 64%, however, it is still drastically worse than our models demonstrate (27.2%) while having just slightly better CER (27.7% vs. 30.0%) on `MathSpeech` benchmark, as shown in Table 4. We should note that

Table 5: `S2L-Sentences` results. Disjoint split. "A": artificially (TTS) annotated audio; "H": human-annotated audio; "Mix": combination of "A" and "H". CER is calculated for lower-case. "Q-$\alpha$B" and "Q-math-$\alpha$B" stand for Qwen2.5-$\alpha$B-instruct and Qwen2.5-math-$\alpha$B-instruct, respectively. "Sent." stands for sentence: metric calculated over the whole sentence; "Eq": only for the embedded equations; "Text": only for the text parts of the sentence.

| Model | Train | Test: H | | | | Test: A | | | |
|---|---|---|---|---|---|---|---|---|---|
| | | CER | | | TeXBLEU | CER | | | TeXBLEU |
| | | Sent. | Text | Eq. | Eq. | Sent. | Text | Eq. | Eq. |
| Q-0.5B | A | 33.74 | 28.23 | 63.44 | 82.99 | 34.34 | 28.89 | 67.38 | 79.58 |
| Q-0.5B | H | 29.18 | 23.13 | 56.93 | 83.22 | 32.69 | 27.05 | 63.48 | 78.55 |
| Q-0.5B | Mix | 32.50 | 28.83 | 54.07 | 83.90 | 32.75 | 29.29 | 53.47 | 79.89 |
| Q-0.5B (25 shots) | H | 30.67 | 27.30 | 63.44 | 73.93 | 31.47 | 27.37 | 72.45 | 66.36 |
| Q-1.5B | A | 33.28 | 27.91 | 59.34 | 84.37 | 32.32 | 27.59 | 56.88 | 80.09 |
| Q-1.5B | H | 25.96 | 20.69 | 53.13 | 83.79 | 27.43 | 22.36 | 55.57 | 78.90 |
| Q-1.5B | Mix | 33.57 | 28.27 | 58.41 | 84.58 | 33.95 | 29.57 | 58.07 | 80.29 |
| Q-1.5B (5 shots) | H | 27.94 | 20.50 | 63.99 | 76.78 | 28.70 | 23.69 | 64.74 | 70.83 |
| Q-1.5B (25 shots) | H | 24.05 | 17.26 | 56.77 | 78.57 | 24.49 | 18.93 | 57.61 | 73.43 |
| Q-math-1.5B | A | 29.49 | 23.56 | 54.72 | 85.05 | 30.35 | 24.24 | 58.48 | 80.18 |
| Q-math-1.5B | H | 23.78 | 18.80 | 45.48 | 85.34 | 27.01 | 22.25 | 51.32 | 79.67 |
| Q-math-1.5B | Mix | 32.16 | 26.79 | 57.83 | 84.36 | 32.40 | 27.36 | 59.56 | 79.88 |
| Q-7B (LoRa) | A | 20.11 | 13.52 | 47.12 | 85.90 | 20.54 | 15.09 | 45.92 | 81.81 |
| Q-7B (LoRa) | H | 20.72 | 14.67 | 46.10 | 84.12 | 22.55 | 16.66 | 50.06 | 79.99 |
| Q-7B (LoRa) | Mix | 18.75 | 12.36 | 43.75 | 85.46 | 19.09 | 13.80 | 43.04 | 81.49 |
| Q-7B (5 shots) | H | 24.19 | 17.56 | 57.91 | 77.97 | 23.44 | 18.76 | 55.88 | 73.31 |
| Q-7B (25 shots) | H | 20.00 | 14.23 | 47.12 | 80.64 | 20.73 | 16.38 | 48.21 | 75.14 |
| SALMONN-13B | A | 23.00 | 15.05 | 57.69 | 82.62 | 17.89 | 10.99 | 49.92 | 80.31 |
| SALMONN-13B | H | 16.03 | 10.09 | 41.53 | 84.62 | 19.68 | 13.60 | 49.69 | 79.34 |
| SALMONN-13B | Mix | 15.43 | 9.57 | 39.68 | 85.76 | 16.78 | 10.42 | 44.96 | 81.48 |

MathSpeech has only 120M parameters, but it was trained on 6-8 million samples. In contrast, our model has 0.5B parameters but was tuned on $\approx$550k samples.

## 5.2 S2L-SENTENCES RESULTS

The results are presented in Table 5 (full version is in Appendix C). We observe that the best performance, in terms of the CER metric, is usually achieved when the model is fine-tuned on human-annotated data. This holds true for both the human-annotated test split and the artificial test split. However, the addition of synthetic data can also benefit both equation-related and text-related metrics. Compared to equation-only conversion, performance on `S2L-sentences` equations' part is noticeably lower. This highlights the added difficulty of transcribing mathematical expressions embedded in context. 5-shot and 25-shot few-shot prompted models perform significantly worse than fine-tuned models of any size. Only Qwen2.5-7B (25 prompts) reaches a comparable result on a human-annotated test (equation part) of 47%. Among all models, the SALMONN Audio-LLM achieves the lowest equation CER of 39.7%, likely due to its large parameter count and its end-to-end design, which reduces dependency on intermediate ASR quality. In contrast to `S2L-equations`, fine-tuned with LoRA Qwen2.5-7B noticeably outperforms smaller models.

## 5.3 DISCUSSION AND LIMITATIONS

To bridge the gap with practical applications, mathematical sentences (`S2L-sentences`) and human annotation were incorporated. However, our data does not fully capture real-world lecture conditions, where equations may be paraphrased, incomplete, or tied to visual content. Addressing this would require costly, fine-grained annotation of lecture recordings, which remains out of scope.

Our work has several limitations. While S2L results are promising, they remain limited in scope and robustness. Post-processing LLMs often fail when ASR transcriptions are vague, and similarly, Audio-LLMs struggle with unfamiliar audio domains. More diverse training data is likely needed. Additionally, while synthetic data is helpful for augmentation, it remains less effective than human speech due to its lower complexity and variability.

## 6 CONCLUSION

In this paper, we introduced `S2L`, a novel large-scale open-source dataset for Speech-to-LaTeX conversion, consisting of 66k human-annotated and 571k TTS-generated audio samples of equations and sentences in English and Russian. Our data collection pipeline is openly described and can support future efforts in speech-driven mathematical understanding. We proposed and evaluated multiple approaches, including ASR post-correction and multimodal end-to-end models. Our models achieved competitive results, outperforming prior work and highlighting the feasibility of S2L conversion when supported by high-quality data. We also demonstrated that handling equations embedded within natural language is substantially more challenging than converting isolated equations. Future work might be devoted to enhancing the dataset with more comprehensive human-annotated real-world data, such as lecture recordings, and improving the conversion quality, with the possible application of audio-visual methods.

## 7 LLM USAGE STATEMENT

LLMs and tools (Grammarly) were used only for grammar correction, text polishing, and shortening.

## 8 ACKNOWLEDGMENT

This work was partially supported by the Ministry of Economic Development of the Russian Federation (Agreement No. 139-10-2025-034 dated June 19, 2025, unique identifier IGK 000000C313925P4D0002). Dmitrii Korzh and Oleg Y. Rogov acknowledge support by grant provided by the Analytical Center in accordance with the subsidy agreement (ID 25-303-64737-2-0017-000001).

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

# A APPENDIX

## A.1 DATASET STATISTICS AND EXAMPLES

Human annotation was performed using the TagMe data labeling platform. For `S2L-equations`, the mixed (Human+Artificial annotation, Eng + Rus) test set includes 2.88k samples, while the train set has 143k samples (+400k Eng MathBridge samples for the "full" mix). The train set comprises 18k human and 53k artificial Russian audio samples, and 21.7k human and 50k artificial English audio samples (+400k artificial English samples). The test set has 54% human-annotated audio, with all test equations distinct from the train set. For `S2L-sentences`, the mixed (Human+Artificial annotation, Eng only) test set contains 2.85k samples, and the train set has 89k samples. Both the train and test sets include 27% human-annotated audio samples.

The `S2L-sentences` dataset has the following stratification by formula length (Table 6) and by the number of equations per sentence (Table 7).

Table 6: Character statistics in `S2L-sentences` dataset for the unique human-annotated expressions.

| Eq. length | 3–10 | 10–20 | 20–30 | 30–50 | 50+ |
|---|---|---|---|---|---|
| **Counter** | 2,752 | 2,751 | 2,552 | 2,396 | 1,941 |

Table 7: Equations per sentence statistics in `S2L-sentences` for the unique human-annotated sentences.

| # Eq. | 1 | 2 | 3 | 4 | 5 | 6–11 |
|---|---|---|---|---|---|---|
| **Counter** | 4,899 | 3,726 | 1,983 | 1,028 | 439 | 321 |

An overview of the dataset collection pipeline is presented in Figure 2.

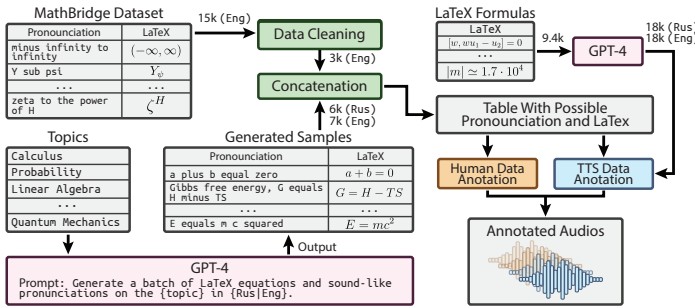

Figure 2: `S2L-equations` collection and annotation pipeline overview.

Let us compare transcriptions of 5 ASR models for one particular human-annotated audio in Table 8.

Let us present several English samples we collected using GPT-4 requests in Table 9. The "Posssible Pronounciation" is necessary for the TTS models to generate speech and is extremely helpful for the human speech annotators as they can use it for reference if they do not know how to read the equation properly and simplifies the criteria for the human annotator.

For the `S2L-sentences`, let us illustrate the evaluation challenge. Consider the CER between the predicted sequence "Given a fixed graph $F$, a typical problem on a large graph $G$ on $n$ vertices that contains no copy of $F$ can have an upper bound on the number of its edges, denoted by $X(n, F)$" and the ground-truth "Given a fixed graph $F$, a typical problem in extremal graph theory asks for the maximum number of edges that a large graph $G$ on $n$ vertices containing no copy of $F$ can have, denoted by $\text{ex}(n, F)$." The equation-only CER is 27.27%.

Table 8: Example of transcription of one particular human-annotated audio of the $\nabla_\nu A^\mu = \frac{\partial A^\mu}{\partial x^\nu} + \Gamma^\mu_{\nu\rho} A^\rho$ equation.

| Model | Transcription |
|---|---|
| Whisper-L | The covariant derivative of a vector a mu equals partial mu with respect to X nu plus gamma upper rho mu nu times a rho. |
| WavLM | the covarient derivative of a vector a mou equals partial moo with respect to ex new plus gama upper row moo new times a row |
| Wav2Vec2 | the covariant derivative of a vector a mu equals partial mo with respect to x-new plus scamma upper row mo new times a row |
| Qwen-audio | The covariant derivative of a vector mu equals partial mu with respect to x nu plus gamma upper row mu nu times a row |
| Canary | The covariant derivative of a vector amu equals partial amu with respect to x nu plus gamma upper rho moon nu times a rho. |

## A.2 METRICS DESCRIPTION

We proceed by examining the primary and additional metrics in detail.

Character Error Rate (CER) which is defined as the ratio of the normalized edit distance (Levenshtein distance) between the predicted sequence and the ground truth, normalized by the total number of characters in the reference:

$$\text{CER} = \frac{S + D + I}{N}, \tag{1}$$

where $S$ is the number of substitutions, $D$ is the number of deletions, $I$ is the number of insertions, and $N$ is the total number of characters in the reference.

The Word Error Rate (WER) is defined similarly to the CER but considers words instead of characters. CER and WER are commonly used in ASR tasks.

ROUGE-1 calculates the unigram recall between the predicted output and the reference text.

$$\text{ROUGE-1} = \frac{\sum_{\text{unigram}\in\text{ref}} \min(\text{count}(\text{unigram}), \text{count}(\text{unigram\_pred}))}{\sum_{\text{unigram}\in\text{ref}} \text{count}(\text{unigram})} \tag{2}$$

This metric is widely used for summarization and transcription tasks to evaluate the lexical overlap between predicted and reference outputs.

BLEU and sacreBLEU evaluate n-gram precision by comparing the predicted output against the reference. BLEU is computed as:

$$\text{BLEU} = \text{BP} \cdot \exp\left(\sum_{n=1}^{N} w_n \log p_n\right) \tag{3}$$

where BP is the brevity penalty, $p_n$ is the precision of n-grams, and $w_n$ are weights. SacreBLEU applies different tokenization (Papineni et al., 2002; Post, 2018). TeXBLEU is a variant of the BLEU score adapted to evaluate LaTeX string generation tasks, particularly mathematical expressions. It penalizes syntactic mistakes and helps measure the quality of generated LaTeX code.

chrF and chrF++ are character-based F-scores metrics that compute a balance between precision and recall at the character level:

$$\text{chrF}_\beta = (1 + \beta^2) \cdot \frac{\text{chrP} \cdot \text{chrR}}{\beta^2 \cdot \text{chrP} + \text{chrR}}, \tag{4}$$

Where chrP and chrR represent the arithmetic mean of character $n$-gram precision and recall across all $n$-grams; chrP is the percentage of character $n$-grams in the hypothesis that also appear in the

Table 9: Example of the dataset samples for further annotation by speaker and TTS models.

| Topic | Possible Pronunciation | Equation |
|---|---|---|
| Calculus. Integrals | Integral: integral of x cubed dx equals x to the fourth over 4 plus constant | $\int x^3 \, dx = \frac{x^4}{4} + C$ |
| Basic Geometry | the distance between two points (x1, y1) and (x2, y2) is the square root of (x2 minus x1) squared plus (y2 minus y1) squared | $d = \sqrt{(x_2 - x_1)^2 + (y_2 - y_1)^2}$ |
| Basic Functions | f of x is equal to x minus 3 divided by x squared minus 9 | $f(x) = \frac{x-3}{x^2-9}$ |
| Partial Derivatives | The partial derivative of f with respect to x and then y equals d squared f divided by d x d y | $\frac{\partial^2 f}{\partial x \partial y}$ |
| Linear Algebra | the cross product of vectors a and b is a vector perpendicular to both | $a \times b$ |
| Differential Equations | the solution to d y over d x equals negative k y is y equals c e to the negative k x | $\frac{dy}{dx} = -ky$ is $y = Ce^{-kx}$ |
| Field Theory | the electromagnetic field tensor is given by F mu nu equals partial mu A nu minus partial nu A mu | $F_{\mu\nu} = \partial_\mu A_\nu - \partial_\nu A_\mu$ |
| Quantum Mechanics | the Schrödinger equation for a free particle is i h bar d psi over d t equals minus h bar squared over 2 m d squared psi over d x squared | $i\hbar\frac{d\psi}{dt} = -\frac{\hbar^2}{2m}\frac{d^2\psi}{dx^2}$ |
| QFT | the Lagrangian density for the gauge field is minus one over four F mu nu F mu nu | $\mathcal{L} = -\frac{1}{4}F_{\mu\nu}F^{\mu\nu}$ |
| Particle Physics | the mass of the Z boson is approximately 91.2 GeV/c squared | $m_Z \approx 91.2 \, \text{GeV}/c^2$ |
| General Physics | Period of a pendulum: two pi times square root of length divided by gravitational acceleration | $T = 2\pi\sqrt{\frac{L}{g}}$ |
| Mathematical Physics | Bessel function of the first kind of order zero, j sub zero is equal to the sum from m equals zero to infinity, of minus one to the power m, divided by m factorial squared, times x divided by two to the power of 2 m | $J_0(x) = \sum_{m=0}^{\infty} \frac{(-1)^m}{(m!)^2}\left(\frac{x}{2}\right)^{2m}$ |
| Trigonometry | Euler's formula, e to the power i times pi plus one equals zero | $e^{i\pi} + 1 = 0$ |
| Thermodynamics | Gibbs free energy, G equals H minus TS | $G = H - TS$ |

reference, and chrR is the percentage of character $n$-grams in the reference that are also found in the hypothesis. chrF++ is chrF for $n = 2$.

TeXBleu is relatively insensitive to the significant errors.

## B  TRAINING HYPERPARAMETERES

The default loss function was cross-entropy, and the default optimizer was AdamW (Loshchilov et al., 2017). Qwen models for `S2L-Equations` experiments were trained on 1 A100 GPUs for 1 epoch, and batch size was set to 16 samples per batch. AdamW optimizer was used with weight decay of $0.01$ with a learning rate $1e-4$ and linear learning rate scheduler.

For `S2L-Sentences` experiments, Qwen models were trained on a single A100 GPU for 1 epoch, and the batch size was set to 16 samples per batch. AdamW optimizer was used with weight decay of $0.01$ with learning rate $1e-4$ and linear learning rate scheduler.

For the Qwen2.5-7B experiments, we applied LoRA with a rank $r = 8$ and a scaling parameter $\alpha = 32$. The adapters were integrated solely into the attention projection matrices.

For the Qwen2-Audio-7B experiments, we used following LoRA configuration ($r = 8$, $\alpha = 16$) targeting the attention projection matrices of the large language model (LLM) and audio encoder backbone and also LLM LMHead.

SALMONN was trained with the LoRA technique on Llama. Target modules were set to attention layers, rank was 8, alpha parameter was 32, and dropout was set to 10%. Whisper and Beats models were frozen. The model was trained on Nvidia H100-80Gb 2 GPUs for 6 epochs. The learning rate was set to 3e-5 with a warm-up for 3000 steps and cosine decay. Gradient accumulation was set to 3 iterations. The batch size was set to 12 samples per batch. Automated mixed precision with float16 was used.

For the few-shot experiments, we employed the pre-trained Qwen2.5-7B Instruct checkpoint without any further fine-tuning.

## C  ADDITIONAL RESULTS

### C.1  ADDITIONAL RESULTS FOR THE MAIN TEXT TABLES

Full version of the Table 2 from the main text for `S2L-equations` results is Table 10. Additional few-show results for the `S2L-sentences` (Table 5) are presented in Table 11.

The few-shot experiments were conducted using pretrained models in instruction mode (in other words, with open-source weights that were not fine-tuned on our dataset by us).

One can notice, that although large end-to-end models demonstrate better perforance, smaller ASR post-sorrection models offer a practical alternative for the resource-constrained environments.

### C.2  ADDITIONAL MODELS

We also evaluated InternLM, ProofGPT, and FlanT5 on the subsets of `S2L-equations` on additional experiments with different splits. Results are presented in Table 12. ProofGPT-1.3B demonstrated good performance, except for the Russian language. In the setting when train and test data both have a mix of genuine and artificial audio, and the test set equations have no overlapping with the equations from the train, SALMONN-13B demonstrates the best metrics except CER on all languages, while Qwen2.5 has a slight edge over SALMONN regarding CER. For instance, on the English subset, SALMONN leads with the highest $\mathrm{Rouge}$-1 (83.88), sBLEU (60.68), and $\mathrm{chrF}$ (71.04) scores. However, its CER (42.42) is slightly higher than Qwen2.5-Math-1.5B, which has the lowest CER (39.54) and ranks second in $\mathrm{Rouge}$-1 (81.43) and $\mathrm{chrF}$ (68.34). The Qwen-Audio performs worse than other methods, probably due to re-implementation nuances. The second part of the table compares Qwen2-0.5B and Qwen2.5-0.5B for English and Russian languages for the random and disjoint (test equations do not overlap train ones) splits. For both languages, Qwen2.5-0.5B consistently outperforms Qwen2-0.5B in terms of $\mathrm{Rouge}$-1 and sBLEU. Interestingly, in the case of the combined English and Russian datasets, the 2 models exhibit very close performance, with Qwen2.5-0.5B showing marginal improvements in accuracy metrics while having a slightly higher CER.

Table 10: `S2L-equations` results. Disjoint split: test equations do not overlap with train equations. "A": artificially (TTS) annotated audio except 400k samples extracted from `MathBridge`; "H": human-annotated audio; "Mix": combination of "A" and "H"; CER is calculated for lowercase. "Q-$\alpha$B" and "Q-math-$\alpha$B" stand for Qwen2.5-$\alpha$B-instruct and Qwen2.5-math-$\alpha$B-instruct, respectively. "Full" implies addition of 400k artificially annotated samples from `MathBridge` to the "A" set.

| Model | Train | Train Language | Test Language | Test: Mix CER | Test: Mix TeXBLEU | Test: H CER | Test: H TeXBLEU | Test: A CER | Test: A TeXBLEU |
|---|---|---|---|---|---|---|---|---|---|
| MathSpeech | MS-train | Eng | Eng | 64.04 | 83.71 | 59.32 | 83.64 | 69.65 | 83.80 |
| Q-0.5B | A | Eng | Eng | 33.28 | 88.61 | 33.26 | 88.54 | 33.30 | 88.70 |
| Q-0.5B | A | Eng+Rus | Eng | 34.78 | 87.90 | 34.94 | 87.57 | 34.59 | 88.31 |
| Q-0.5B | H | Eng | Eng | 36.91 | 87.86 | 35.01 | 88.25 | 39.16 | 87.38 |
| Q-0.5B | H | Eng+Rus | Eng | 35.43 | 88.06 | 33.94 | 88.47 | 37.19 | 87.56 |
| Q-0.5B | Mix | Eng | Eng | 31.41 | 88.83 | 31.06 | 88.87 | 31.82 | 88.78 |
| Q-0.5B | Mix | Eng+Rus | Eng | 32.33 | 88.60 | 31.18 | 88.89 | 33.69 | 88.24 |
| Q-0.5B | Mix-full | Eng+Rus | Eng | 27.21 | 90.20 | 27.03 | 90.14 | 27.42 | 90.27 |
| Q-1.5B | A | Eng | Eng | 31.24 | 89.22 | 31.37 | 89.15 | 31.08 | 89.31 |
| Q-1.5B | A | Eng+Rus | Eng | 30.73 | 88.92 | 30.70 | 88.73 | 30.77 | 89.16 |
| Q-1.5B | H | Eng | Eng | 29.69 | 89.41 | 27.57 | 89.69 | 32.22 | 89.07 |
| Q-1.5B | H | Eng+Rus | Eng | 30.93 | 89.04 | 28.85 | 89.42 | 33.39 | 88.57 |
| Q-1.5B | Mix | Eng | Eng | 29.76 | 89.28 | 28.93 | 89.44 | 30.74 | 89.09 |
| Q-1.5B | Mix | Eng+Rus | Eng | 31.14 | 89.37 | 30.08 | 89.43 | 32.40 | 89.28 |
| Q-1.5B | Mix-full | Eng+Rus | Eng | 25.69 | 90.70 | 24.91 | 90.74 | 26.61 | 90.66 |
| Q-math-1.5B | A | Eng | Eng | 29.44 | 89.61 | 30.00 | 89.33 | 28.77 | 89.96 |
| Q-math-1.5B | A | Eng+Rus | Eng | 29.57 | 90.00 | 29.44 | 89.80 | 29.74 | 90.23 |
| Q-math-1.5B | H | Eng | Eng | 30.16 | 89.83 | 28.97 | 90.13 | 31.58 | 89.46 |
| Q-math-1.5B | H | Eng+Rus | Eng | 31.45 | 89.25 | 30.71 | 89.43 | 32.34 | 89.02 |
| Q-math-1.5B | Mix | Eng | Eng | 28.53 | 89.97 | 28.08 | 90.13 | 29.05 | 89.76 |
| Q-math-1.5B | Mix | Eng+Rus | Eng | 27.75 | 89.85 | 27.54 | 89.89 | 28.01 | 89.79 |
| Q-math-1.5B | Mix-full | Eng+Rus | Eng | 25.01 | 90.90 | 25.05 | 90.90 | 24.97 | 90.89 |
| Q-7B | A | Eng | Eng | 28.15 | 90.10 | 28.07 | 89.96 | 28.25 | 90.26 |
| Q-7B | A | Eng+Rus | Eng | 27.32 | 90.12 | 26.16 | 90.20 | 28.70 | 90.03 |
| Q-7B | H | Eng | Eng | 27.97 | 89.99 | 26.93 | 90.29 | 29.20 | 89.62 |
| Q-7B | H | Eng+Rus | Eng | 26.89 | 90.18 | 26.43 | 90.36 | 27.44 | 89.95 |
| Q-7B | Mix | Eng | Eng | 26.10 | 90.58 | 25.80 | 90.68 | 26.46 | 90.45 |
| Q-7B | Mix | Eng+Rus | Eng | 27.78 | 90.11 | 26.55 | 90.37 | 29.24 | 89.78 |
| Q-7B | Mix-full | Eng+Rus | Eng | 26.17 | 90.50 | 25.96 | 90.51 | 26.43 | 90.48 |
| Qwen-Audio | Mix | Eng | Eng | 71.67 | 83.55 | 104.19 | 78.45 | 33.06 | 89.84 |
| SALMONN | Mix-full | Eng | Eng | 17.50 | 93.68 | 18.17 | 93.64 | 16.70 | 93.72 |
| Gemma 3n | Mix-full | Eng | Eng | 34.24 | 89.15 | 33.24 | 89.23 | 35.42 | 89.06 |
| Q-0.5B | A | Rus | Rus | 32.06 | 91.73 | 40.95 | 94.13 | 27.62 | 90.53 |
| Q-0.5B | A | Eng+Rus | Rus | 9.63 | 95.14 | 17.78 | 96.34 | 5.56 | 94.54 |
| Q-0.5B | H | Rus | Rus | 15.24 | 96.97 | 15.87 | 96.76 | 14.92 | 97.08 |
| Q-0.5B | H | Eng+Rus | Rus | 6.77 | 97.74 | 13.97 | 97.20 | 3.17 | 98.01 |
| Q-0.5B | Mix | Rus | Rus | 15.34 | 98.50 | 15.24 | 97.21 | 15.40 | 99.14 |
| Q-0.5B | Mix | Eng+Rus | Rus | 13.02 | 97.32 | 14.60 | 96.78 | 12.22 | 97.59 |
| Q-0.5B | Mix-full | Eng+Rus | Rus | 8.15 | 97.03 | 15.56 | 96.68 | 4.44 | 97.21 |
| Q-1.5B | A | Rus | Rus | 10.05 | 96.77 | 19.37 | 96.60 | 5.40 | 96.85 |
| Q-1.5B | A | Eng+Rus | Rus | 6.14 | 96.62 | 14.60 | 96.74 | 1.90 | 96.56 |
| Q-1.5B | H | Rus | Rus | 4.66 | 99.38 | 8.25 | 99.49 | 2.86 | 99.33 |
| Q-1.5B | H | Eng+Rus | Rus | 14.50 | 97.38 | 14.60 | 96.67 | 14.44 | 97.73 |
| Q-1.5B | Mix | Rus | Rus | 14.60 | 95.90 | 10.79 | 98.00 | 16.51 | 94.85 |
| Q-1.5B | Mix | Eng+Rus | Rus | 4.02 | 99.20 | 11.75 | 97.68 | 0.16 | 99.96 |
| Q-1.5B | Mix-full | Eng+Rus | Rus | 4.55 | 98.89 | 13.33 | 96.74 | 0.16 | 99.96 |
| Q-math-1.5B | A | Rus | Rus | 6.03 | 98.94 | 17.14 | 97.02 | 0.48 | 99.89 |
| Q-math-1.5B | A | Eng+Rus | Rus | 11.01 | 98.04 | 13.33 | 97.08 | 9.84 | 98.52 |
| Q-math-1.5B | H | Rus | Rus | 13.23 | 96.51 | 2.86 | 99.36 | 18.41 | 95.08 |
| Q-math-1.5B | H | Eng+Rus | Rus | 12.49 | 97.45 | 11.75 | 97.78 | 12.86 | 97.28 |
| Q-math-1.5B | Mix | Eng+Rus | Rus | 5.50 | 98.90 | 13.65 | 97.39 | 1.43 | 99.66 |
| Q-math-1.5B | Mix | Rus | Rus | 17.25 | 97.24 | 12.70 | 97.70 | 19.52 | 97.01 |
| Q-math-1.5B | Mix-full | Eng+Rus | Rus | 13.33 | 97.86 | 14.29 | 96.72 | 12.86 | 98.43 |
| Q-7B | A | Rus | Rus | 3.70 | 99.19 | 10.79 | 97.66 | 0.16 | 99.96 |
| Q-7B | A | Eng+Rus | Rus | 6.14 | 98.39 | 16.83 | 96.46 | 0.79 | 99.35 |
| Q-7B | H | Rus | Rus | 5.40 | 99.29 | 6.98 | 99.33 | 4.60 | 99.27 |
| Q-7B | H | Eng+Rus | Rus | 21.59 | 96.73 | 14.92 | 97.00 | 24.92 | 96.60 |
| Q-7B | Mix | Rus | Rus | 1.59 | 99.57 | 4.44 | 98.78 | 0.16 | 99.96 |
| Q-7B | Mix | Eng+Rus | Rus | 4.66 | 99.09 | 13.65 | 97.36 | 0.16 | 99.96 |
| Q-7B | Mix-full | Eng+Rus | Rus | 7.94 | 97.74 | 14.92 | 96.87 | 4.44 | 98.17 |
| Flamingo 3 | Mix | Rus | Rus | 2.01 | 99.88 | 0.00 | 100.00 | 3.02 | 99.82 |
| SALMONN | Mix-full | Rus | Rus | 9.38 | 97.73 | 6.51 | 99.55 | 10.81 | 96.82 |
| Gemma 3n | Mix-full | Rus | Rus | 15.30 | 97.70 | 11.26 | 98.17 | 17.33 | 94.48 |

## C.3 Rest of the metrics

We tried to train LLM with pronunciations from all 5 ASR systems from Table 8 to make it an ASR-agnostic model, but the model's accuracy was worth more than just with Whisper. For results see Table 13.

Table 11: `S2L-sentences` results for Few-Shot experiments. Disjoint split: test sentences do not overlap with train sentences. "A": artificially (TTS) annotated audio; "H": human-annotated audio; "Mix": combination of "A" and "H". CER is calculated for lower-case. "Q-$\alpha$B" and "Q-math-$\alpha$B" stand for Qwen2.5-$\alpha$B-instruct and Qwen2.5-math-$\alpha$B-instruct, respectively. "Sent." stands for sentence: metric calculated over the whole sentence; "Eq": only for the embedded equations; "Text": only for the text parts of the sentence.

| Model | Train | Test: H | | | | Test: A | | | |
|---|---|---|---|---|---|---|---|---|---|
| | | CER | | | TeXBLEU | CER | | | TeXBLEU |
| | | Sent. | Text | Eq. | Eq. | Sent. | Text | Eq. | Eq. |
| **5-shot** | | | | | | | | | |
| Q-0.5B | A | 35.80 | 35.76 | 66.22 | 66.87 | 34.21 | 34.20 | 68.30 | 60.41 |
| Q-0.5B | H | 32.09 | 29.69 | 71.95 | 65.31 | 31.83 | 30.16 | 73.71 | 58.47 |
| Q-1.5B | A | 26.16 | 20.62 | 58.78 | 76.84 | 26.64 | 22.26 | 60.47 | 70.64 |
| Q-1.5B | H | 27.94 | 20.50 | 63.99 | 76.78 | 28.70 | 23.69 | 64.74 | 70.83 |
| Q-math-1.5B | A | 35.94 | 35.99 | 62.73 | 71.66 | 41.90 | 43.85 | 65.75 | 64.53 |
| Q-math-1.5B | H | 42.52 | 42.16 | 73.36 | 68.23 | 44.63 | 45.15 | 78.05 | 61.07 |
| Q-7B | A | 23.83 | 18.44 | 56.25 | 77.88 | 23.63 | 19.57 | 56.03 | 72.64 |
| Q-7B | H | 24.19 | 17.56 | 57.91 | 77.97 | 23.44 | 18.76 | 55.88 | 73.31 |
| | | | | | | | | | |
| **25-shot** | | | | | | | | | |
| Q-0.5B | A | 28.74 | 25.97 | 61.87 | 73.77 | 28.15 | 25.89 | 65.14 | 66.53 |
| Q-0.5B | H | 30.67 | 27.30 | 63.44 | 73.93 | 31.47 | 27.37 | 72.45 | 66.36 |
| Q-1.5B | A | 23.65 | 17.55 | 56.84 | 78.42 | 24.10 | 18.82 | 58.77 | 72.39 |
| Q-1.5B | H | 24.05 | 17.26 | 56.77 | 78.57 | 24.49 | 18.93 | 57.61 | 73.43 |
| Q-math-1.5B | A | 37.65 | 29.11 | 88.22 | 75.14 | 36.74 | 27.56 | 95.89 | 69.09 |
| Q-math-1.5B | H | 30.58 | 24.43 | 67.93 | 75.67 | 31.26 | 27.51 | 67.43 | 68.44 |
| Q-7B | A | 21.22 | 15.85 | 50.43 | 79.88 | 21.65 | 16.97 | 51.97 | 74.65 |
| Q-7B | H | 20.00 | 14.23 | 47.12 | 80.64 | 20.73 | 16.38 | 48.21 | 75.14 |

We measured case-sensitive performance (for example, $\phi$ and $\Phi$ mean different symbols). Results are presented in Tables 14 and 15. As we can see, the performance drop is not as severe. This generally means that models were trained well and that data regarding capitalized and non-capitalized symbols was labelled well. The rest of the metrics from the Table 12 are represented in Table 16 as an addition with the lower-cased metrics for the `S2L-eqautions` part.

## C.4 CROSS-LANGUAGE LEARNING.

One of the advantages of fine-tuning multilingual language models is the ability to extract information from one language that is not available in another. For example, LaTeX special symbols `\simeq` and `\hat` are not presented in the Russian part of the equations dataset but in English. Qwen2.5, trained in English and Russian, can transcribe "approximately equal" in Russian to `\simeq` ($\simeq$). Another observation is that the models are primarily English-oriented, so Qwen2.5-Math-1.5B and Qwen2-0.5B trained in Russian can generate only simple formulas in English. The reverse situation works worse - Qwen2.5-0.5B, trained in English, cannot perform post-correction in Russian.

## C.5 ADDITIONAL ERROR ANALYSIS

Let us present error analysis on `S2L-equations` test. Among 2.8k equations, around 1.8k predictions are not exactly identical to the LaTeX reference, but more than half of these mismatches have a character overlap above 0.8, indicating that the majority of errors are local rather than structural.

- The main part of errors is based on symbol substitutions or index errors. Some of the issues arise from ASR errors. For example:
  - pred `y_{1},y_{2},y_{3} = (i,s,1)`,
    true `(y_{1},y_{2},y_{3}) = (\phi,\psi,1)`

Table 12: `S2L-equations` (subset) results. SALMONN represent end-to-end Audio-LLMs, while all other models use ASR post-correction via fine-tuned LLMs. "A" denotes artificially (TTS) annotated audio, "H" refers to human-annotated audio, and "Mix" indicates a combination of both. "Rand" indicates a random split where equation-pronunciation-speaker/voice triplets are non-overlapping across train, validation, and test sets. "Disj" specifies a disjoint split where test equations do not appear in the training set.

| Model | Lang | Train | Test | Split | CER↓ | Rouge-1↑ | sBLEU↑ | chrF↑ |
|---|---|---|---|---|---|---|---|---|
| Qwen2.5-0.5B | Eng | Mix | Mix | Disj | 43.87 | 77.78 | 53.33 | 64.48 |
| Qwen2.5-Math-1.5B | Eng | Mix | Mix | Disj | **39.54** | 81.43 | 57.86 | 68.34 |
| ProofGPT-1.3B | Eng | Mix | Mix | Disj | 41.60 | 78.04 | 52.31 | 64.30 |
| InternLM2-1.8B | Eng | Mix | Mix | Disj | 49.23 | 78.12 | 61.00 | 64.24 |
| Flan-T5 | Eng | Mix | Mix | Disj | 64.92 | 53.47 | 11.98 | 28.78 |
| SALMONN-13B | Eng | Mix | Mix | Disj | 42.42 | **83.88** | **60.68** | **71.04** |
| Qwen2.5-0.5B | Rus | Mix | Mix | Disj | 13.19 | 89.71 | 72.78 | 86.09 |
| Qwen2.5-Math-1.5B | Rus | Mix | Mix | Disj | 10.49 | 90.66 | 74.25 | 88.11 |
| ProofGPT-1.3B | Rus | Mix | Mix | Disj | 16.48 | 87.82 | 70.82 | 84.04 |
| SALMONN-13B | Rus | Mix | Mix | Disj | **10.45** | 93.59 | 76.63 | 91.63 |
| Qwen2.5-0.5B | Eng+Rus | Mix | Mix | Disj | **22.70** | 86.22 | 67.14 | 79.87 |
| ProofGPT-1.3B | Eng+Rus | Mix | Mix | Disj | 23.93 | 84.85 | 65.33 | 78.18 |
| SALMONN-13B | Eng+Rus | Mix | Mix | Disj | 24.27 | **89.93** | **69.62** | **84.10** |
| Qwen2-0.5B | Eng | A | H | Rand | 25.05 | 86.56 | 70.39 | 76.91 |
| Qwen2.5-0.5B | Eng | A | H | Rand | **23.56** | **86.92** | **71.37** | **77.88** |
| Qwen2-0.5B | Rus | A | H | Rand | **7.09** | 94.44 | 79.59 | **92.79** |
| Qwen2.5-0.5B | Rus | A | H | Rand | 7.49 | **94.58** | **79.88** | 92.73 |
| Qwen2-0.5B | Eng+Rus | A | H | Disj | **30.36** | 83.52 | 61.72 | 72.20 |
| Qwen2.5-0.5B | Eng+Rus | A | H | Disj | 31.13 | **83.60** | **61.73** | **72.22** |

Table 13: `S2L-equations` (subset). Metrics results (%) for Qwen trained with 5 ASR models.

| Model | CER↓ | Rouge-1↑ | sBLEU↑ | chrF↑ | WER↓ | METEOR↑ | BLEU↑ | chrF++↑ |
|---|---|---|---|---|---|---|---|---|
| Qwen2.5-0.5B | 43.21 | 78.49 | 50.06 | 60.35 | 75.33 | 57.21 | 47.06 | 58.88 |

Table 14: `S2L-equations` (subset). Case-sensitive metrics (%) for different Language Models. "Mix" means a combination of human-annotated and TTS. Lang means the language of the train/validation/test splits.

| Model | Lang | Train | Test | Split | CER↓ | Rouge-1↑ | sBLEU↑ | chrF↑ |
|---|---|---|---|---|---|---|---|---|
| Qwen2.5-0.5B | Eng | Mix | Mix | Disj | 45.79 | 77.78 | 50.46 | 61.06 |
| Qwen2.5-Math-1.5B | Eng | Mix | Mix | Disj | **44.39** | 79.29 | 51.02 | 61.67 |
| SALMONN-13B | Eng | Mix | Mix | Disj | 44.47 | **83.88** | **56.76** | **66.70** |
| Flan-T5 | Eng | Mix | Mix | Disj | 67.52 | 53.47 | 10.43 | 26.01 |
| Qwen-Audio | Eng | Mix | Mix | Disj | 54.64 | 76.63 | 54.79 | 57.61 |
| Qwen2.5-0.5B | Rus | Mix | Mix | Disj | 13.45 | 89.71 | 72.67 | 85.47 |
| SALMONN-13B | Rus | Mix | Mix | Disj | **10.59** | **93.59** | **76.52** | **91.38** |
| Qwen2.5-0.5B | Eng+Rus | Mix | Mix | Disj | **23.39** | 86.22 | 66.26 | 78.74 |
| SALMONN-13B | Eng+Rus | Mix | Mix | Disj | 24.99 | **89.93** | **68.69** | **82.82** |

  – pred `\frac{\partial\mathcal L}{\partial\phi}`,
    true `\frac{\partial\L}{\partial\phi}`.
- A significant part of errors involves missing or ambiguous `\text{...}` blocks.

  – pred: `f(x)=\lambda e^{-\lambda x}, x\ge 0`,
    true: `f(x)=\lambda e^{-\lambda x}\text{for }x\ge 0.`

Table 15: `S2L-equations` (subset). Remaining case-sensitive metrics (%) for different Language Models. "Mix" means combination of Human annotated and TTS. Lang means language of train/validation/test splits

| Model | Lang | Train | Test | Split | WER↓ | METEOR↑ | BLEU↑ | chrF++↑ |
|---|---|---|---|---|---|---|---|---|
| Qwen2.5-0.5B | Eng | Mix | Mix | Disj | 79.60 | 56.89 | 47.16 | 59.44 |
| Qwen2.5-Math-1.5B | Eng | Mix | Mix | Disj | 76.78 | 57.52 | 47.85 | 60.24 |
| SALMONN-13B | Eng | Mix | Mix | Disj | **72.20** | **61.91** | **53.08** | **65.06** |
| Flan-T5 | Eng | Mix | Mix | Disj | 111.83 | 20.47 | 6.19 | 24.84 |
| Qwen-Audio | Eng | Mix | Mix | Disj | 102.91 | 53.67 | 42.53 | 55.89 |
| Qwen2.5-0.5B | Rus | Mix | Mix | Disj | 28.14 | 80.78 | 70.55 | 83.68 |
| SALMONN-13B | Rus | Mix | Mix | Disj | **18.13** | **84.91** | **74.95** | **90.09** |
| Qwen2.5-0.5B | Eng+Rus | Mix | Mix | Disj | 42.46 | 73.63 | 63.80 | 78.18 |
| SALMONN-13B | Eng+Rus | Mix | Mix | Disj | **40.02** | **77.24** | **66.77** | **81.38** |

Table 16: `S2L-equations` (subset). Remaining results of lower-case metrics (%) for different models. SALMONN represents the Multimodal approach, while the rest of the models represent ASR post-correction. "A" stands for artificially annotated audio (TTS), "H" – human annotated audio, "Mix" – the combination of both "A" and "H". "Disj" split means that test equations do not intersect with the train equations, and "Rand" split means that train-test split was made randomly over generated pairs and equations from train might occur in the test but should be pronounced with different speakers or TTS models.

| Model | Lang | Train | Test | Split | WER↓ | METEOR↑ | BLEU↑ | chrF++↑ |
|---|---|---|---|---|---|---|---|---|
| Qwen2.5-0.5B | Eng | Mix | Mix | Disj | 76.85 | 56.89 | 50.42 | 62.71 |
| Qwen2.5-Math-1.5B | Eng | Mix | Mix | Disj | 69.16 | 60.33 | 55.57 | 66.77 |
| ProofGPT-1.3B | Eng | Mix | Mix | Disj | 69.64 | 55.86 | 49.73 | 62.50 |
| SALMONN-13B | Eng | Mix | Mix | Disj | **68.90** | **61.91** | **57.55** | **69.20** |
| InternLM2-1.8B | Eng | Mix | Mix | Disj | 81.01 | 57.30 | 50.65 | 62.55 |
| Flan-T5 | Eng | Mix | Mix | Disj | 109.26 | 20.47 | 7.69 | 27.53 |
| Qwen2.5-0.5B | Rus | Mix | Mix | Disj | 27.14 | 80.78 | 70.64 | 84.34 |
| Qwen2.5-Math-1.5B | Rus | Mix | Mix | Disj | 23.80 | 81.65 | 72.03 | 86.47 |
| ProofGPT-1.3B | Rus | Mix | Mix | Disj | 32.14 | 79.10 | 68.51 | 82.22 |
| SALMONN-13B | Rus | Mix | Mix | Disj | **17.94** | **84.91** | **75.05** | **90.36** |
| Qwen2.5-0.5B | Eng+Rus | Mix | Mix | Disj | 41.47 | 73.63 | 64.75 | 78.18 |
| ProofGPT-1.3B | Eng+Rus | Mix | Mix | Disj | 43.26 | 72.20 | 62.94 | 76.37 |
| SALMONN -13B | Eng+Rus | Mix | Mix | Disj | **38.80** | **77.24** | **67.85** | **82.62** |
| Qwen2-0.5B | Rus | A | H | Rand | 14.82 | 86.74 | 78.46 | 91.87 |
| Qwen2.5-0.5B | Rus | A | H | Rand | **13.91** | **86.77** | **78.77** | **91.92** |
| Qwen2-0.5B | Eng | A | H | Rand | 40.37 | 73.88 | 68.60 | 76.53 |
| Qwen2.5-0.5B | Eng | A | H | Rand | **38.54** | **74.59** | **69.71** | 76.53 |
| Qwen2-0.5B | Eng+Rus | A | H | Disj | **57.02** | **68.83** | **58.82** | 70.78 |
| Qwen2.5-0.5B | Eng+Rus | A | H | Disj | 58.27 | 68.56 | 58.60 | **70.85** |

  – pred: `(P^{e})^{\theta},`
  true: `(p^{e})^{\text{th}}.`
  – pred: `\sin(x) \quad\Rightarrow\quad y...,`
  true: `\sin(x) \text{ is } y = ...`
- Fractions and nested fractions.
  – pred: `\frac{\delta\mathcal{L}}{\delta(\partial_\mu\phi)} = 0,`
  true: `frac{\partial\L}{\partial(d\phi/dx^{\mu})} = 0.`

Overall, the majority of differences between predictions and labels are small lexical or index-level discrepancies, as well as occasional text–math boundary issues. Structural failures, such as incorrect integrals or deeply nested fraction constructions, are uncommon.

