# OpenReview forum: "Speech-to-LaTeX: New Models and Datasets for Converting Spoken Equations and Sentences"
_ICLR.cc/2026/Conference — ICLR 2026 Poster_

### Official Review · Reviewer_fXZ7 · 2025-10-17

**Soundness:** 3
**Presentation:** 3
**Contribution:** 2
**Rating:** 4
**Confidence:** 5

**Summary:**

The paper introduces the first large-scale open-source dataset of spoken mathematical expressions and sentences in English and Russian, containing about 66,000 human-annotated audio samples from diverse scientific domains. It also provides evaluations of several speech-to-latex methods, including ASR post-correction, few-shot prompting, and audio-LLM integration, showing improvements over existing baselines such as MathSpeech.

Overall, this is a useful dataset paper with clear execution but limited novelty and a somewhat narrow scope for ICLR.

**Strengths:**

The work contributes a valuable resource for an understudied task and offers solid baselines that could support future research. The dataset appears well-organized and the experiments are clearly presented.

**Weaknesses:**

However, the topic is quite niche and seems better aligned with speech or audio processing venues such as Interspeech or ICASSP rather than ICLR. The scientific originality is limited, the main contribution is dataset creation and benchmarking rbut for a very nich domain.

**Questions:**

Some details need clarification, including which crowdsourcing platform was used for human annotation and the ratio of synthetic to natural speech samples in the dataset created. Providing these is important for the dataset publication.

---

> ### Author Response · Authors · 2025-11-20
>
> Dear reviewer,
>
> Thank you for your valuable feedback. We are pleased to hear that you found our work to be useful, the dataset well-organized, and the experiments presented clearly. Let us address your questions and concerns.
>
>
> $\textbf{...the topic is quite niche and seems better aligned with speech or audio processing venues...}$
>
> It is possible to reach an agreement that a speech-processing conference might be more suitable for our work. However, our work was submitted to the ''datasets and benchmarks'' area, which covers a broad range of benchmarks and experimental results, and well aligns with the primary contribution of our article. For instance, one can find that the following works devoted to the audio and multimodal domain have been accepted in previous years in the ''datasets and benchmarks'' area:
>
> - Shah, Muhammad A., et al. "Speech Robust Bench: A Robustness Benchmark For Speech Recognition."
>
> - Zhang, Hanlei, et al. "MIntRec2. 0: A Large-scale Benchmark Dataset for Multimodal Intent Recognition and Out-of-scope Detection in Conversations."
>
> In addition, the MathSpeech article was accepted to another similar A* conference.
>
> Moreover, the applicability of the dataset is not limited to the mathematical expression recognition tasks, as it can also be used for general ASR, voice biometrics, and voice anti-spoofing, expanding existing datasets.
>
>
> $\textbf{The scientific originality is limited, the main contribution is dataset creation and benchmarking ...}$
>
> The second contribution is the result of the recognition of mathematical expressions and the first result of the conversion of the mathematical sentence, both of which, as discussed in this manuscript, are positive and adequate. Therefore, one can attempt to propose more original and sophisticated methods in future studies, even for wider-scope tasks.
>
>
>
>
> $\textbf{...which crowdsourcing platform was used for human annotation ...}$
>
> Human annotation was performed using the TagMe data labeling platform. We have added this information to the Appendix of the revised version of the manuscript.
>
>
>
> $\textbf{...the ratio of synthetic to natural speech samples in the dataset created...}$
>
> It can be estimated from information from subsection 3.3, particularly from lines 229 to 241, where statistics on utterance per subset, language, and type of generation are described.  Summing up, for the $\texttt{S2L-sentences}$, there is a 67k/24.8k $\approx 2.7/1$ ratio, and for the $\texttt{S2L-eqautions}$, including 400k arifiically annotated samples from MathBridge, there is 504k/41k $\approx 12.3/1 $. However, $\texttt{S2L-sentences}$ test set has 27\% human-annotated sentences and $\texttt{S2L-equations}$ test set has 54\% human-annotated expressions.
>
>
>
>
> We hope that our response properly addresses the questions and concerns raised by the reviewer. And we hope that the reviewer will consider increasing the score of our paper. We are looking forward to addressing any remaining concerns.

---

> > ### Comment · Reviewer_fXZ7 · 2025-11-21
> > **response acknowledgement**
> >
> > tks for your answer
> > knowing that the paper was specifically submitted to the ''datasets and benchmarks'' track, i will increase my score

---

> > > ### Author Response · Authors · 2025-12-03
> > >
> > > Thank you for your reply

---

### Official Review · Reviewer_Sqnu · 2025-10-28

**Soundness:** 3
**Presentation:** 3
**Contribution:** 2
**Rating:** 6
**Confidence:** 4

**Summary:**

In this paper, the author proposes a new benchmark dataset for the speech-to-LaTeX transcription task. Compared with the existing benchmark, MathSpeech, the proposed dataset includes a broader range of samples that better reflect real-world speech-to-text usage. It introduces more challenging cases, such as mixed text and mathematical formulas, making the benchmark more representative of practical scenarios. In addition, unlike MathSpeech, which relies on synthetic speech, this dataset is collected from real human speakers, further enhancing its realism and applicability. Using the proposed dataset, the author trains models with two different processing approaches, following established speech-to-LaTeX training pipelines, and benchmarks their performance to evaluate the effectiveness of current algorithms in this direction.

**Strengths:**

1. The benchmark dataset proposed in this paper covers a wider range of diverse, real-world scenarios, making it a valuable contribution to the research community.
2. The paper provides a comprehensive exploration of the dataset’s potential applications, including post-transcription correction and end-to-end multimodal fine-tuning.
3. The dataset-splitting strategy is well designed, and the experimental results effectively demonstrate the generalization ability of different models when adapting to various languages and previously unseen formula patterns.

**Weaknesses:**

1. The technical novelty of the paper is limited. The work mainly reimplements common approaches to fine-tuning models without proposing new model architectures or training algorithms to address the issues identified in the experiments.
2. The Character Error Rate (CER) remains relatively high in many cases, and the paper lacks an in-depth analysis of the causes behind these failures. A more thorough investigation into the unexpected phenomena observed in the results would strengthen the contribution.
3. Several parts of the paper are not well organized. For instance, some sections—such as the discussion of OCR methods in the related works—contain redundant content. Additionally, Table 2 is placed far from the results section, making it inconvenient for readers to reference while reviewing the experimental findings.

**Questions:**

1. For the few-shot learning experiments, were they conducted on a fine-tuned model or directly on the original pretrained model?
2. Could the authors provide more details about the data split ratios used in different experimental settings (e.g., training vs. testing)? For instance:
    - What is the ratio of training to testing equations in the Disjoint setting?
    - What are the training and testing sample sizes under the source-type split?
    - In the monolingual and bilingual experiments, do you use all available examples from one language plus a portion of the other language for training? If so, please specify the ratio used.

---

> ### Author Response · Authors · 2025-11-20
>
> Dear reviewer,
>
> Thank you for your valuable feedback. Let us address your questions and concerns.
>
>
> $\textbf{W1.}$
>
> Indeed, we submitted the article to the ''datasets and benchmarks'' part of the conference, as our primary contribution is the benchmarks. However, the second contribution is based on the successful recognition of mathematical expression sentences. As discussed in this manuscript, the results are positive. Therefore, there is an opportunity to develop more innovative and advanced methods in future research.
>
> $\textbf{W2.}$
>
> This concern is highlighted in several parts of the manuscripts, including the introduction (lines 81-83), subsection 4.3, subsection 5.1 (lines 406-408 submitted version, lines 407-409  in updated version), and Table 3. The primary reason is evaluation ambiguity in mathematical expressions, even for the TeXBLEU. The second reason is errors in ASR predictions, feature encoding variation of ASR submodels, and domain shift in both LLM and Audio-LM models. The third reason (in mathematical sentences) is context handling. In general, adding more data and diverse samples improves performance.
>
> $\textbf{W3.}$
>
> OCR LaTeX recognition methods are discussed on purpose, as our dataset contains equations obtained (from the labels) of the OCR $\texttt{TextTeller}$ dataset. Please refer to lines 183-186 in 3.1.
>
>
> $\textbf{Q1.}$
>
> The few-shot experiments were conducted using pretrained models in instruction mode (in other words, with open-source weights that were not fine-tuned on our dataset by us). We have added this information to the Appendix of the revised version of the text.
>
>
>
>
> $\textbf{Q2.}$
>
> For $\texttt{S2L-equations}$, the mixed (Human+Artificial annotation, Eng + Rus) test set includes 2.88k samples, while the train set has 143k samples (+400k Eng MathBridge samples for the ''Mix-full''). The train set comprises 18k human and 53k artificial Russian audio samples, and 21.7k human and 50k artificial English audio samples (+400k artificial English samples). The test set has 54\% human-annotated audio, with all test equations distinct from the train set.
>
> For $\texttt{S2L-sentences}$, the mixed (Human+Artificial annotation, Eng only) test set contains 2.85k samples, and the train set has 89k samples. Both the train and test sets include 27\% human-annotated audio samples.
>
>
> We have added clarification in the revised manuscript. For some additional details, please refer to subsections 3.1 and 3.3.
>
>
> We hope that this response addresses the reviewer's questions and concerns. We also hope that the reviewer will consider raising the scores of our paper. We are eager to address any remaining issues.

---

### Official Review · Reviewer_rLuj · 2025-11-01

**Soundness:** 3
**Presentation:** 3
**Contribution:** 3
**Rating:** 6
**Confidence:** 3

**Summary:**

The paper introduces Speech-to-LaTeX (S2L)—the large-scale, open dataset for converting spoken math into LaTeX—addressing limits of prior work that used only synthetic audio, required two ASR passes, and handled only isolated equations. S2L includes 66k human and 571k TTS samples of equations and sentences in English/Russian. Two solutions are evaluated: (1) Whisper + Qwen post-correction, and (2) end-to-end audio-LLMs (e.g., SALMONN). On S2L-equations, Qwen reaches ~25–30% CER, and SALMONN achieves 17.5%, far outperforming MathSpeech (~64% on S2L). The paper also establishes the first benchmark for spoken mathematical sentences, where multimodal models remain strongest but accuracy drops due to contextual ambiguity.

**Strengths:**

- **Large, open S2L resource:** Releases a two-part dataset (S2L-equations, S2L-sentences) with multilingual coverage (English/Russian), mixing 66k human and 571k synthetic clips, collected from diverse sources and 33 annotators. This addresses the data bottleneck and standardizes evaluation.
- **Clear task framing & thorough splits:** Uses disjoint-formula splits, human vs. TTS source splits, and mono vs. bilingual training setups—plus KaTeX-based equation normalization—to probe generalization beyond memorization.
- **Strong baselines across paradigms:** Compares post-ASR pipelines (Qwen 0.5B/1.5B/Math/7B) with end-to-end Audio-LLMs (SALMONN, Flamingo-3, Gemma-3n, Qwen-Audio), giving a balanced picture of modular vs. E2E.
- **Transparent MathSpeech comparison:** After normalization, MathSpeech degrades to 64% CER on S2L-equations, while the proposed models remain strong; also competitive on the original MathSpeech benchmark.
- **Practical touches in evaluation:** High KaTeX compile rate (≈98–99.5%) suggests many errors are minor formatting differences rather than catastrophic syntax failures.

**Weaknesses:**

- **Compute/latency not discussed:** The best model (SALMONN-13B) is likely heavy; no throughput/latency/memory reporting limits practical takeaways for real-time use.
- **Ambiguity handling is under-analyzed:** The work acknowledges inherent ambiguity (“one over x plus two”) but gives limited breakdowns by ambiguity type or guidance on disambiguating conventions during annotation/evaluation.
- **Model behavior anomalies lack ablations:** 7B (LoRA-tuned, frozen base) underperforms fully fine-tuned 1.5B on equations; Qwen-Audio “fails completely.” Causes are hypothesized but not probed via targeted ablations.
- **Multilingual training instability:** Adding Russian sometimes harms English performance; analysis is brief (imbalance) without deeper diagnosis of cross-lingual interference.
- **Real-world robustness is unclear:** Data is largely clean/read speech; the gap to noisy, disfluent lecture audio (and multimodal cues, e.g., slides/pointing) isn’t evaluated.
- **Tokenization tweak had no effect:** Adding special LaTeX tokens didn’t help; this is noted but not further explored (e.g., tokenizer training, subword strategies).
- **Metric–semantics gap:** Heavy reliance on CER/TeXBLEU—even with normalization—can penalize semantically equivalent LaTeX or underplay mathematically serious mistakes; no semantic/visual-equivalence metric is reported.

**Questions:**

**Practicality & generalization**

- What are end-to-end latencies and memory footprints for the strongest systems, and can lighter E2E models approach SALMONN’s accuracy?
- How robust are models to real-world conditions (noise, disfluencies, accents, mic distance)? Any stress-tests or augmentation plans?
- Multilingual effects: can you quantify negative transfer vs. language imbalance and test mitigation (balanced sampling, language tags, adapters)?
- Fairness of comparisons: would training a MathSpeech-style architecture on S2L (or retraining Qwen on MathSpeech’s 6–8M) clarify the role of data vs. model size?

**Dataset & evaluation**

- How were inherently ambiguous utterances annotated (e.g., explicit “parentheses” vs. natural speech)? Any inter-annotator agreement stats and adjudication policies for these cases?
- Could you add a semantic check (e.g., render-and-compare, AST comparison) alongside CER/TeXBLEU to better reflect user-perceived correctness?
- Do you have a typology of common error modes (nested fractions, integrals, Greek variants, boundaries between text and math) for equations-in-sentences?
- Can you release the KaTeX normalization code and exact split scripts to facilitate strict reproducibility?

**Modeling choices & ablations**

- What ablations isolate why 7B (LoRA) < 1.5B (full FT) on equations? Is it LoRA rank, target layers, or optimizer/schedule?
- Why did Qwen-Audio fail? Can you share a brief post-mortem (feature pipeline, adapter alignment, loss setup)?
- Did adding 400k MathBridge-derived TTS samples change error composition (e.g., more bracket errors decreased), or only average CER?
- Given that Qwen-Math didn’t clearly beat Qwen on this task (inputs are NL), did math-specialized token priors help particular symbol families or structures?

---

> ### Author Response · Authors · 2025-11-20
> **Part 1**
>
> Dear reviewer,
>
> Thank you for your valuable feedback. Let us address your questions and concerns.
>
> $\textbf{W1.}$ Even though the SALMONN model is 13B and requires about 27-30 GB VRAM, it still achieves a Real-time factor (RTF) of about 0.1758 on H100 GPU with mixed precision.
>
>
> $\textbf{W2.}$ We acknowledge the need for deeper ambiguity analysis. While semantic ambiguities were discussed in Section 5.3 as unresolvable from audio, we addressed syntactic ambiguities via normalization (Sections 3.2, 4.3).
>
> $\textbf{W3.}$ We apologize for the inconvenience. In fact, the 7B model often outperforms other smaller LLMs and does not always underperform 1.5B or 0.5B models. For instance, let us refer to Table~10 in the appendix. For the English human annotated subpart of the $\texttt{S2L-equations}$ testset, Qwen-7B (Q-7B) slightly outperforms smaller models in all scenarios, except for the ''Mix-full'' train set. However, for the Russian human annotated test subpart,  Q-7B demonstrates better results only for the following train set splits: (A, Rus), (Mix, Rus). We have rewritten this statement in the revised version of the text.
>
> We conducted preliminary ablation studies on the 7B model. Brief results:
>
> - Increasing LoRA rank did not yield performance improvements.
>
> - Partial unfreezing strategy (lm\_head and the last two layers) underperformed compared to the LoRA approach.
>
> $\textbf{English Only}$. Train mix, test mix (both have artificial and human-annotated audio). Disjoint split.
>
> - Qwen2.5-7B (Lora r=8, a=32): CER=26.10
>
> - Qwen2.5-7B (Lora r=16, a=64): CER=27.39
>
> - Qwen2.5-7B (Unfreeze the LMHead and the last two layers.): CER=34.22
>
>
>
> We also considered similar ablations for Qwen2-Audio-7B models and achieved similar results, that increasing LoRA rank did not yield performance improvements.
>
> $\textbf{English Only}$. Train mix, test mix (have both artificial and human-annotated audio). Disjoint split.
>
> - Qwen2-Audio-7B (Lora r=8, a=16): CER=71.67
>
> - Qwen2-Audio-7B (Lora r=8, a=32): CER=72.84
>
> - Qwen2-Audio-7B (Lora r=16, a=64): CER=72.21
>
>
> $\textbf{W4.}$ We acknowledge that cross-lingual training requires deeper diagnosis in future research. Some discussion is presented in Appendix C.3.
> However, the effect might be explained from cross-lingual interference perspectives, not just due to data imbalance. The addition of the second language might have a positive regularization meaning, but can also lead to cross-lingual interference. Both of these effects might worsen performance on the considered language, but balancing them might improve the results, for instance, in the case of well-balanced data and training strategy, regularization should prevail. Thus, as the Eng dataset is larger than Rus, the impact of the addition of Rus data to Eng is smaller than in the vice-versa scenario, in which results might be changed significantly for the Rus test.
>
> $\textbf{W5.}$  We agree that robust performance on noisy, disfluent speech is crucial for real-world applications. This is a key objective for our future research, as noted in the limitations and in the conclusion. Audio augmentation during training, such as the addition of noise and reverberation, can partially improve the empirical robustness of the models.
>
> $\textbf{W6.}$ Modern LLMs are trained on a large plethora of data, which is several magnitudes larger than our dataset. Tokenizer can handle special symbols efficiently, even though it does not have separate tokens for them through context learning. Moreover, LLMs have the majority of tokens of relevant Latex symbols.  Thus, fine-tuning the tokenizer may result in weak improvements or even slight underperformance. Nonetheless, this finding suggests diving into tokenization strategies, which we plan to explore in future work.

---

> ### Author Response · Authors · 2025-11-20
> **Part 2**
>
> $\textbf{W7.}$ We recognize the limitations of CER/TeXBLEU for capturing semantic equivalence, as we discussed it in several parts of the text. Automatic visual equivalence metrics/AST-based similarity metrics also have similar semantic understanding disadvantages and require the same equation normalization/standardization techniques as for the CER. For instance, consider $\verb|1/2|$, $\verb|0.5|$, $\verb|\frac{1}{2}|$, all of which would be rendered differently and have different abstract syntax trees. For example, edit distance between AST of $\verb|\frac{1}{2} + x^2|$ and $\verb|1/2 + x^2|$ is 3. Additionally, $\verb|a+b|$ and $\verb|b+a|$ have the same semantics but different representation (however, this example is less representative for our data, as the order of arguments is usually preserved in annotation). More fair manual assessment of equivalence is labour-intensive.  However, this is an important direction for future research, and we would appreciate any advice.
>
> We have computed edit distances between ASTs to evaluate semantic equivalence. The results on the mixed (human and artificial) multilingual training and English-only test splits are as follows:
>
>
> - Qwen2.5-0.5B AST Edit Dist = 4.18 CER = 32.33
>
> - Qwen2.5-1.5B AST Edit Dist = 4.04, CER = 31.14
>
> - Qwen2.5-7B AST Edit Dist = 3.44, CER = 27.78
>
> - MathSpeech AST Edit Dist = 6.113, CER = 64.04
>
>
> These results demonstrate a clear correlation between the AST-based metric and CER.
>
> $\textbf{Practicality and Generalization}$
>
> $\textbf{Q1.}$ Please refer to our response to W1 and to Tables 2, 5, 10 in the text. The best performance-latency trade-off might be achieved with 1.5B models. In the case of equation recognition, even 0.5B models might be considered as suitable. Even though the SALMONN model is 13B and requires about 27-30 GB VRAM, it still achieves a Real-time factor (RTF) of about 0.1758 on H100 GPU with mixed precision.
>
> $\textbf{Q2.}$ The robustness of our ASR post-correction baselines is inherently limited by the underlying Whisper model's capabilities, which we did not modify. Robustness of end-to-end approaches also partially relies on the robustness of audio encoders (for instance, one of SALMONN's audio encoders is Whisper), and investigating the robustness of the SALMONN-based approach to such real-world conditions is a valuable direction for future work. Whisper is well-known to be robust to various noises and sound domains, and further robustness improvements of post-correction and multimodal end-to-end approaches might be achieved through augmentations (noises, distortions, room impulse responses) and additional data incorporation. However, the second audio encoder of SALMONN, which is focused on music encoding, is more susceptible to noise and reverberation.
>
>
>
> $\textbf{Q3.}$ Balance sampling not improves the results neither for ''Mix'' split (32.91 vs. 32.33 Q-0.5B Eng) not for the ''Mix-full'' split (29.67 vs.27.21).
>
> $\textbf{Q4.}$ Unfortunately, MathSpeech does not provide training code and train dataset. It only provides test benchmark and inference code, which lacks the essential part of its approach: hypothesis correction prior to conversion to LaTeX. Additionally, we do not have the computational budget to artificially annotate 8 million samples in a reasonable time in order to replicate MathSpeech's experiments with Qwen models. Nonetheless, we tried to re-implement the MathSpeech approach, training one t5 model for textual error correction and another t5 model for the LaTeX conversion. For the checkpoint trained with t5-small initialization, we obtained CER=50.62\% on the English mix annotated test of $\texttt{S2L-equations}$, which is better than the performance of their $\texttt{MathSpeech}$-trained checkpoint (64.0\%, see Table 4) but still worse than our Qwen-0.5B (27.2\%). Data vs. model size influence can be partially inherited from the results of the tables and might be considered as a prominent future research direction.
>
> $\textbf{Dataset Evaluation}$
>
> $\textbf{Q5.}$ Human annotators were provided with no strict rules on how to pronounce ambiguous expressions, except for being understandable and valid. From our observations, parentheses are mainly omitted in speech and reference pronunciations, the subscript is usually spoken/written directly, and the superscript (power) is usually said after the subscript and may not be specifically highlighted.
>
>
>
> $\textbf{Q6.}$ Please, refer to the comment on W7.

---

> ### Author Response · Authors · 2025-11-20
> **Part 3**
>
> $\textbf{Q7.}$
> Let us present error analysis on $\texttt{S2L-equations}$ test.
> Among 2.8k equations, around 1.8k predictions are not exactly identical to the
> LaTeX reference, but more than half of these mismatches have a character
> overlap above 0.8, indicating that the majority of errors are local rather than
> structural.
>
>
>
> The main part of errors is based on symbol substitutions or index errors. Some of the issues arise from ASR errors. For example:
>
> - pred  $\verb| y_{1},y_{2},y_{3} = (i,s,1)|$, true  $\verb|(y_{1},y_{2},y_{3}) = (\phi,\psi,1).|$
>
> - pred  $\verb|\frac{\partial\mathcal L}{\partial\phi}|$, true  $\verb|\frac{\partial\L}{\partial\phi}|$
>
>
> A significant part of errors involves missing or ambiguous $\verb|\text{...}|$ blocks.
>
>     - pred: $\verb|f(x)=\lambda e^{-\lambda x}, x\ge 0|$, true: $\verb|f(x)=\lambda e^{-\lambda x}\text{for }x\ge 0|$
>     - pred: $\verb|(P^{e})^{\theta}|$, true: $\verb|(p^{e})^{\text{th}}|$
>     - pred: $\verb|\sin(x) \quad\Rightarrow\quad y = ...|$, true: $\verb|\sin(x) \text{ is } y = ...|$
>
>
> Fractions and nested fractions.
>
> -  pred: $\verb|\frac{d}{dx^\mu}\frac{\delta\mathcal{L}}{\delta(\partial_\mu\phi)} = 0|$, true: $\verb|\frac{d}{dx^{\mu}}\frac{\partial\L}{\partial(d\phi/dx^{\mu})} = 0|$
>
>
>
> Overall, the majority of differences between predictions and
> labels are small lexical or index-level discrepancies, as well as occasional
> text--math boundary issues. Structural failures, such as incorrect integrals or
> deeply nested fraction constructions, are uncommon. We added a concise
> discussion of these error modes to the appendix.
>
>
> $\textbf{Q8.}$ All source code will be publicly available. The dataset is fully and anonymously available (no authors, affiliations) on a well-known platform and can be easily found. However, to avoid indirect breach of anonymity or of any other regulations, we will refrain from providing a link (as, at least, there was a non-anonymous external contributor commit). The dataset is organized into disjoint train and test sets (with additional required metadata such as annotation type and language). At present, we propose to refer to the supplementary materials.
>
>
> $\textbf{Modeling Choices and Ablations}$
>
> $\textbf{Q9.}$  Please refer to the comment on W3.
>
>
>
> $\textbf{Q10.}$ Unfortunately, Qwen-Audio has no official fine-tuning code or guide (but has inference). Source code for the Qwen-Audio training is available in the supplementary materials. Please refer to the comment on W3 for additional details.
>
>
>
> $\textbf{Q11.}$ We did not analyze it deeply, as we did not obtain unexpected results: CER was improved for the English subset of the test, for the Russian subset of the test result became worse the more English data was in the train (Table 10), and the compilation success rate remained similar.
>
>
> $\textbf{Q12.}$ We have not performed a fine-grained analysis to determine if the math-specialized token priors in Qwen-Math provided targeted benefits. This remains an open question for future investigation.
>
>
>
> We trust that the following response adequately addresses the reviewer's inquiries and concerns. We would also appreciate the reviewer's consideration in revising the scores attributed to our manuscript. We remain committed to addressing any remaining issues that may arise.

---

### Official Review · Reviewer_1KKx · 2025-11-10

**Soundness:** 2
**Presentation:** 3
**Contribution:** 2
**Rating:** 4
**Confidence:** 4

**Summary:**

The paper addresses Speech-to-LaTeX for both isolated equations and full sentences, releasing a large bilingual dataset with human and TTS audio. It compares Whisper with LLM post-correction against end-to-end Audio-LLMs like SALMONN-13B, showing strong gains over prior works (MathSpeech).

**Strengths:**

1) This paper targets Speech-to-LaTeX, an underexplored but impactful task for education and research with large-scale S2L dataset.

2) It shows strong empirical results on S2L-equations (English). SALMONN achieves 17.5% CER, outperforming MathSpeech and Qwen; on S2L-sentences, SALMONN attains the best equation CER (39.7%).

**Weaknesses:**

1) Gap to real lecture conditions. Authors note the dataset does not capture paraphrases, incomplete expressions, or audio-visual coupling typical of classroom settings.

2) It is difficult to verify the reliability of the dataset presented in the paper. Although some sample data are available in the supplementary material, those samples are insufficient to establish whether the constructed dataset adequately covers diverse scenarios of spoken mathematical expressions.

3) Reproducibility risk for one baseline. Qwen-Audio “fails completely,” attributed to re-implementation differences—suggesting pipeline brittleness.

**Questions:**

1) Could you quantify how much equation normalization (and lowercasing) changes CER/TeXBLEU per model? Also, any experiments with structure-aware or render-equivalence metrics? Concrete numbers would help judge true semantic progress.

2) Did you try constrained decoding (grammar or AST constraints) in the post-correction step to reduce bracket/scope errors? If so, how did it affect latency and CER?

3) This paper states, ‘We release the first large-scale, open-source dataset of spoken mathematical expressions and sentences.’ Is there an anonymized data pages related to this?

---

> ### Author Response · Authors · 2025-11-20
>
> Dear reviewer,
>
> Thank you for your valuable feedback. Let us address your questions and concerns.
>
>
> $\textbf{W1.}$
>
> Indeed, this concern is addressed in the limitations section (5.3, lines 458-460) and is suggested as a potential area for future research in the conclusion.
>
> $\textbf{W2.}$
>
> The dataset coverage is discussed in several parts of the manuscript, i.e., in Section 3, particularly in 3.1 ''Data Sources and Preparation'' and primarily in 3.4 ''Data Representatives Discussion''.
> Moreover, in addition to the ''sample\_dataset'' folder with audio samples, there are .csv tables containing much more ground truth expressions, pronunciation, predictions, and metrics in the supplementary. These can be found in the ''source\_code/multimodal/Salmonn/results'' folder. Additionally, please refer to the answer to Question 3.
>
>
> $\textbf{W3.}$
> There is no reproducibility risk for the Qwen-Audio as we stated bad performance rather than great. Vice versa, one has a chance to obtain good results with Qwen-Audio :)
>
> In any case, if you are aware of a valid and reliable open-source fine-tuning framework for Qwen-Audio, we would be sincerely grateful if you could share it with us. We will promptly try to conduct experiments using this code.
>
>
> For the Qwen-Audio-7B experiments, we used LoRA configuration $r = 8$, $\alpha \in \{16, 32\}$ and $r=16$, $\alpha = 64$ targeting the attention projection matrices, MLPs and LMHead of the large language model (LLM) backbone and audio encoder.
>
> The following results were achieved for the mix train and test sets. $\textbf{English Only}$. Train mix, test mix (have both artificial and human-annotated audio). Disjoint split.
>
> - Qwen2-Audio-7B (Lora r=8, a=16): CER=71.67
>
> - Qwen2-Audio-7B (Lora r=8, a=32): CER=72.84
>
> - Qwen2-Audio-7B (Lora r=16, a=64): CER=72.21
>
> Qwen-Audio performance is significantly worse than the ASR post-correction baselines with typical CER about 30.
>
>
> $\textbf{Q1.}$
>
>
> $\textbf{Case sensitivity}$
>
> Let us compare the CER metric (in \%, lowercase v.s. case sensitive) using results of additional experiments on a subsample of train data (no Mathbridge artificial audio, around 40k training audio, see Table 12 and Table 14).
>
> English only. Train mix, test mix (have both artificial and human-annotated audio). Disjoint split.
>
> - Qwen2.5-0.5B: 43.87 vs 45.79
>
> - Qwen2.5-Math-1.5B: 39.54 vs 44.39
>
> - Salmonn-13B: 42.42 vs 44.47
>
>
> Russian only. Train mix, test mix (have both artificial and human-annotated audio). Disjoint split.
>
> - Qwen2.5-0.5B: 13.19 vs 13.45
>
> - Salmonn-13B: 10.45 vs 10.49
>
> One can notice that it is slightly harder to predict case-sensitive equations.
>
> $\textbf{Normalization Effect}$
>
> Let us compare the CER metric (in \%, with normalization v.s. without normalization) using the main dataset without Mathbridge samples:
>
> English only. Train mix, test mix (have both artificial and human-annotated audio). Disjoint split.
>
> - Qwen2.5-0.5B: 31.41 vs 34.95
>
> - Qwen2.5-1.5B: 29.76 vs 34.68
>
> - Qwen2.5-Math-1.5B: 29.77 vs 34.20
>
>
> As we can see, normalization reduces the error rate by 3.5 to 4.5 percentage points.
>
> $\textbf{Q2.}$
>
> No, we did not. The latex compilation error rate was relatively low (up to 99\% compilation success rate), but thank you for the suggestion.
>
>
> $\textbf{Q3.}$
>
> The dataset is fully and anonymously available (no authors, affiliations) on a well-known platform and can be easily found. However, to avoid indirect breach of anonymity or of any other regulations, we will refrain from providing a link (as, at least, there was a non-anonymous external contributor commit).
>
> We trust that our response has adequately addressed the reviewer's queries and concerns. We also extend our hope that the reviewer will reconsider and potentially increase the score of our paper. We remain committed to resolving any remaining issues.

---

### Meta-Review · Area_Chair_69ya · 2026-01-07

**Summary:**

All reviewers have acknowledged the utility of the data set and the strong empirical results. There are some concerns about generalization to real-word scenarios (raised by reviewer 1KKx and rLuj) and other minor issues, such as how ambiguities are handled (raised by rLuj) and whether the technical novelty is limited (raised by reviewer Sqnu). Overall, the data set is valuable and the experimental methodology is sound.

**Reviewer Concerns:**

The generalization to real-word scenarios remains a problem. Even though this is raised by both reviewer 1KKx and rLuj, the contribution of the paper has been fairly clear, and the problem can be left as future work.

**Reviewer Scores:**

Reviewer fXZ7 has explicitly agreed to raise the score.

---

### Decision · Program_Chairs · 2026-01-26

Accept (Poster)